# *Bombyx* Vasa sequesters transposon mRNAs in nuage via phase separation requiring RNA binding and self-association

Hiroya Yamazaki[1], Yurika Namba[1], Shogo Kuriyama[1], Kazumichi M. Nishida[1], Asako Kajiya[1,2] & Mikiko C. Siomi [1] ✉

*Bombyx* Vasa (BmVasa) assembles non-membranous organelle, nuage or Vasa bodies, in germ cells, known as the center for Siwi-dependent transposon silencing and concomitant Ago3-piRISC biogenesis. However, details of the body assembly remain unclear. Here, we show that the N-terminal intrinsically disordered region (N-IDR) and RNA helicase domain of BmVasa are responsible for self-association and RNA binding, respectively, but N-IDR is also required for full RNA-binding activity. Both domains are essential for Vasa body assembly in vivo and droplet formation in vitro via phase separation. FAST-iCLIP reveals that BmVasa preferentially binds transposon mRNAs. Loss of Siwi function derepresses transposons but has marginal effects on BmVasa-RNA binding. This study shows that BmVasa assembles nuage by phase separation via its ability to self-associate and bind newly exported transposon mRNAs. This unique property of BmVasa allows transposon mRNAs to be sequestered and enriched in nuage, resulting in effective Siwi-dependent transposon repression and Ago3-piRISC biogenesis.

Vasa, an ATP-dependent RNA helicase of the DEAD (Asp-Glu-Ala-Asp)-box protein family, was originally identified in *Drosophila* as a factor essential for the development of the female reproductive system[1,2]. Subsequent studies showed that Vasa is highly conserved in a wide range of organisms including humans and accordingly serves as a hallmark of primordial and gonadal germ cells[3,4].

In developing embryos of *Drosophila*, Vasa localizes to polar granules, where it specifically binds to some mRNAs and regulates translation, embryonic patterning, and germ cell differentiation[5,6]. Vasa in germ cells is detected in nuage, a non-membranous perinuclear organelle[7–10]. Nuage is known to act as the biogenesis center of PIWI-interacting RNA (piRNA)-induced silencing complex (piRISC)[11,12]. The piRISC is composed of piRNA and its protein partner PIWI in a stoichiometric manner, and plays a central role in piRISC-mediated transposon silencing[13–18]. piRISC generation, particularly its amplification process known as the ping-pong cycle, takes place within nuage, where two PIWI members [e.g., Aubergine (Aub) and Ago3 in

*Drosophila* and Mili and Miwi2 in mice] cleave transposon transcripts complementary to PIWI-bound piRNAs depending on their endonuclease (slicer) activity. This system accumulates numerous piRISCs in germ cells and concomitantly consumes transposon transcripts in both sense and antisense orientations, achieving transposon silencing at the post-transcriptional level[14,19,20]. *Vasa* mutant flies do not have nuage and so contain few piRISCs[21]. As a result, transposons are desilenced as in *piwi* mutants[22]. This causes severe defects in gonadal development, resulting in infertility.

Further analysis showed that Vasa of *Drosophila melanogaster* (hereinafter referred to as DmVasa) binds to piRNA precursors exported from the nucleus in a UAP56-dependent manner and assembles nuage to trigger the ping-pong cycle[23]. (Notably, a recent study showed that piRNA precursors are also exported in an Nxf3-dependent manner in the gonads[24], although DmVasa's involvement in this pathway remains unclear.) This model is consistent with the earlier notion that DmVasa is an RNA-binding protein and is necessary for the

[1]Department of Biological Sciences, Graduate School of Science, The University of Tokyo, Tokyo 113-0032, Japan. [2]Present address: Department of Medical Chemistry, Graduate School of Medicine, Kyoto University, Kyoto 606-8501, Japan. ✉e-mail: siomim@bs.s.u-tokyo.ac.jp

formation of nuage[25]. However, the mechanism behind nuage formation has remained unclear. It has also remained unclear how other properties of DmVasa, such as RNA unwinding[26], contribute to piRISC generation in *Drosophila*.

Studies using *Bombyx* ovary-derived cultured germ cells (e.g., BmN4 cells) revealed that *Bombyx mori* Vasa (BmVasa) uses its own RNA unwinding activity to release cleaved RNAs from Siwi-piRISC (Siwi is the counterpart of *Drosophila* Aub)[11,27]. The E339Q mutant of BmVasa remains bound to hydrolyzed ATP (i.e., ADP and inorganic phosphate) and is therefore unable to release RNAs (Glu339 is the second amino acid in the DEAD box of BmVasa)[11,27]. In addition, the E339Q mutant remains attached to Siwi-piRISC and piRNA-free Ago3. These results suggested a model of the role of BmVasa in the ping-pong cycle; namely, BmVasa unwinds cleaved RNAs from Siwi-piRISC to promote the transfer to unbound Ago3 by interacting with the two PIWI proteins simultaneously. Interestingly, BmVasa is not involved in RNA release from Ago3-piRISC in the ping-pong cycle. We recently identified another DEAD-box protein, Ddx43, as a factor responsible for the process of promoting the generation of Siwi-piRISC depending on Ago3 slicer activity[28]. Factors other than DEAD-box proteins involved in piRISC biogenesis in silkworm include mitochondrial proteins, such as Papi and Zucchini (Zuc)[29,30].

The nuage in BmN4 cells can be classified into at least three types: Ago3 bodies, Vasa bodies, and Mael bodies[31]. Ago3 bodies function as centers for generating Siwi-piRISC in the ping-pong cycle, while Vasa bodies are thought to function as centers for generating Ago3-piRISC. The function of Mael bodies in piRISC biogenesis remains unknown. Upon depletion of Ago3 in BmN4 cells, Ago3 bodies are no longer assembled, but Vasa bodies are still present. Similarly, the loss of Mael results in the loss of Mael bodies, but the formation of Vasa bodies is unaffected. These findings suggest that the formation of Vasa bodies is at the top of the hierarchy of nuage formation[31].

Human Ddx4/Vasa (Ddx4) localizes to nuage in both ovaries and testes[32]. Ddx4 contains intrinsically disordered regions (IDRs) at both N- and C-termini in addition to the central DEAD-box RNA helicase domain[33]. When Ddx4 was ectopically expressed in non-gonadal HeLa cells, which intrinsically express no Ddx4 and contain no nuage or piRISC, liquid–liquid phase separation (LLPS) resulted in the formation of nuage-like structures[33]. The Ddx4[YFP] mutant, which has a yellow fluorescent protein (YFP) instead of the central RNA helicase domain, also assembled the droplets, albeit in the nucleus. The N-IDR alone also formed nuage-like droplets in vitro but was destabilized by amino acid alterations or modifications that disrupt electrostatic interaction of N-IDRs. These findings indicated that the N-IDR of Ddx4 is necessary and sufficient for LLPS-driven droplet formation, but that the RNA-binding activity via the RNA helicase domain is not required. However, it is still unclear whether this idea applies to Vasa in other species, including BmVasa.

In this study, we analyzed BmVasa in cultured BmN4 cells and found that the N-IDR and the central RNA helicase domain are responsible for self-association and RNA binding, respectively, and that the RNA-binding activity of the RNA helicase domain is originally negligible but is greatly enhanced by self-association via the N-IDR. The activities of BmVasa to both self-associate (via the N-IDR) and bind RNAs (via the central RNA helicase domain) were indispensable for Vasa body formation. Recombinant BmVasa phase-separated and assembled nuage-like droplets in vitro, which was enhanced in the presence of single-stranded RNAs (ssRNAs). Fully automated and standardized individual-nucleotide-resolution cross-linking immunoprecipitation (FAST-iCLIP) sequencing for BmVasa showed that BmVasa preferentially binds transposon mRNAs, the primary target of Siwi-mediated transposon silencing and the major source of Ago3-piRISC. Unlike transposon expression, BmVasa–RNA binding was little affected by the lack of Siwi-piRISC, suggesting that BmVasa targets and binds to RNAs that have just been exported from the nucleus. These

findings indicate that BmVasa assembles nuage (Vasa bodies) via phase separation depending on its ability to self-associate and bind preferentially to newly exported transposon mRNAs. This property of BmVasa allows transposon mRNAs to be sequestered and enriched in Vasa bodies, leading to effective Siwi-dependent post-transcriptional repression of transposons and concomitant Ago3-piRISC biogenesis.

## Results

### Both N-IDR and central RNA helicase domain are required for BmVasa to assemble Vasa bodies and facilitate Ago3-piRISC generation in BmN4 cells

BmVasa, like its human ortholog Ddx4, contains two IDRs, one at the N-terminal end (N-IDR: Met1–Val135) and the other at the C-terminal end (C-IDR: Gly565–Trp601) (Fig. 1a and Supplementary Fig. 1a). The N-IDR of Ddx4 (Met1–Ser236) was reported to be required and sufficient for the self-association of Ddx4 and formation of nuage-like structures[33] in HeLa cells. To test whether the N-IDR of BmVasa is also required and sufficient for the self-association of BmVasa and assembly of Vasa-positive nuage (i.e., Vasa bodies) in BmN4 germ cells, we produced two mutants of BmVasa, ΔN and ΔC, that lack N-IDR and C-IDR, respectively (Fig. 1a). WT BmVasa and the deletion mutants were expressed individually in BmN4 cells, and immunoprecipitation and immunofluorescence were conducted.

Immunoprecipitation showed that Flag-BmVasa WT and the ΔC mutant co-immunoprecipitated with Myc-BmVasa WT, but Flag-BmVasa ΔN mutant did not (Fig. 1a). Immunofluorescence showed that Flag-BmVasa WT and its ΔC mutant localized to Vasa bodies in BmN4 cells but Flag-BmVasa ΔN mutant failed to do so (Fig. 1b and Supplementary Fig. 1b). In BmN4 cells lacking endogenous BmVasa, where Vasa bodies were no longer assembled[31], expression of Flag-BmVasa WT and the ΔC mutant restored the nuage assembly (Fig. 1c). However, no such result was obtained with the ΔN mutant. These findings indicate that the N-IDR of BmVasa is essential for self-association of BmVasa, BmVasa localization to Vasa bodies, and de novo assembly of Vasa bodies in BmN4 cells. The C-IDR of BmVasa was not required for any of these activities.

We next examined whether the BmVasa ΔN and ΔC mutants retained the ability to produce Ago3-piRISC. Upon depletion of endogenous BmVasa, WT BmVasa and the mutants were expressed and piRNAs loaded onto Flag-Ago3, which was co-expressed in the cells, were visualized by [32]P-labeling. When WT BmVasa and the ΔC mutant were expressed, Flag-Ago3 was similarly loaded with piRNAs (Fig. 1d and Supplementary Fig. 1c). However, when the ΔN mutant was expressed, Flag-Ago3 was almost empty as in the negative control cells. These results indicate that BmVasa N-IDR is indispensable for the production of Ago3-piRISC. Ago3 was dispersed throughout the cytoplasm in BmVasa ΔN-expressing cells (Fig. 1e). Because Ago3 body formation requires Ago3 being loaded with piRNAs[34], this result corroborates that BmVasa N-IDR is indispensable for the generation of Ago3-piRISC.

We produced another mutant of BmVasa, termed N-EGFP-C, containing an EGFP instead of the central RNA helicase domain, mimicking the Ddx4[YFP] mutant[33], and examined its subcellular localization in BmN4 cells. The N-EGFP-C mutant did not localize to Vasa bodies when expressed in normal BmN4 cells (Fig. 1f and Supplementary Fig. 1b). It also failed to assemble Vasa bodies in endogenous BmVasa-lacking cells (Fig. 1f), unlike the Ddx4[YFP] mutant in HeLa cells. This clearly indicates that the N-IDR of BmVasa is not sufficient for Vasa body assembly in BmN4 cells and that the central RNA helicase domain is additionally required for BmVasa to induce the body formation. The contribution of N-IDR to Vasa/Ddx4 function appears to differ between silkworm and humans.

The central RNA helicase domain in BmVasa and Ddx4, which consists of a DEAD-like helicase N-terminal domain (DEXDc) and a helicase superfamily C-terminal domain (HELICc), is conserved in

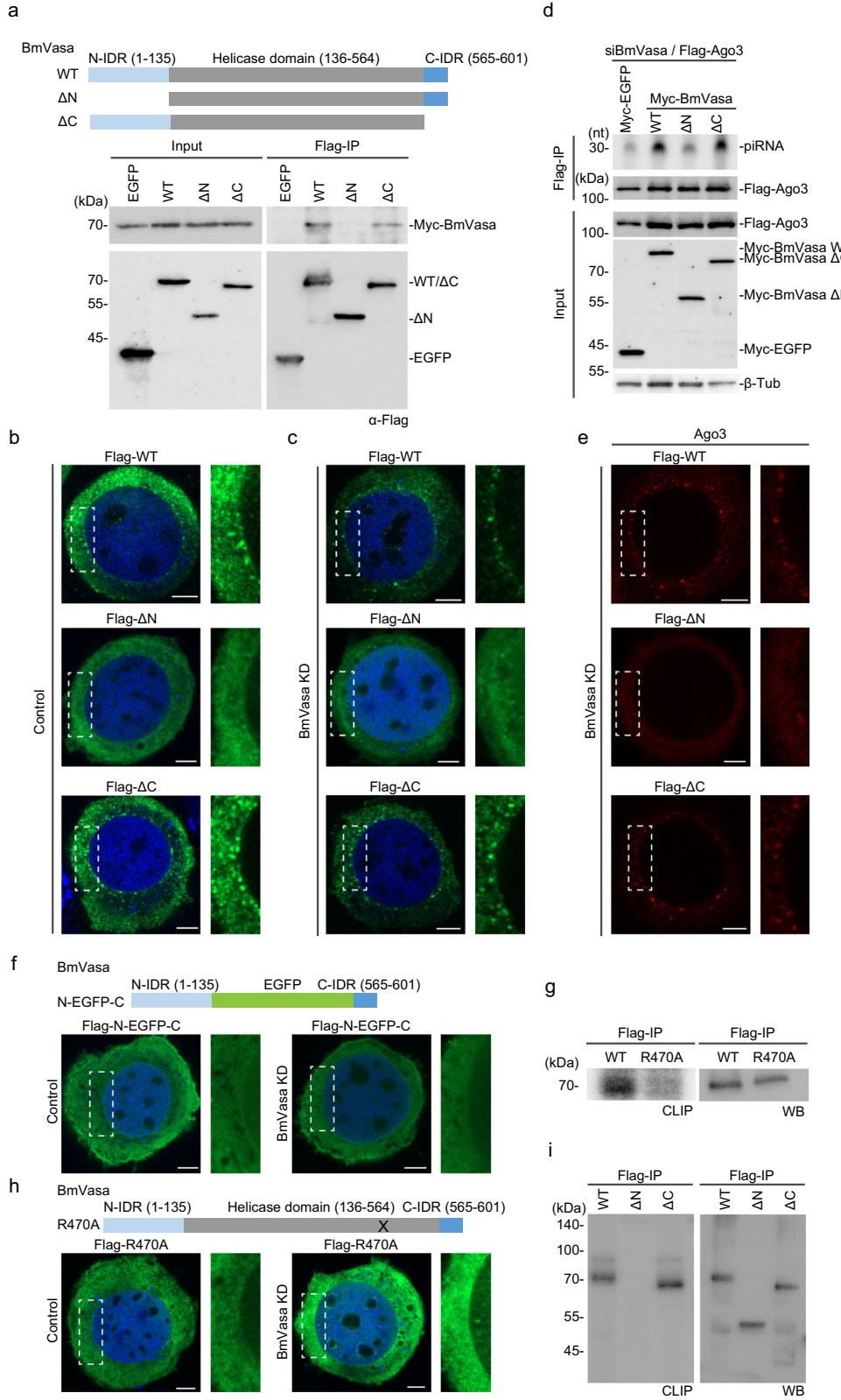

many factors functioning in RNA metabolism and confers RNA-binding activity to host proteins[35]. To exert its function, the RNA helicase domain must contain a Gln-X-X-Arg (QXXR) tetrad in HELICc[36]. This tetrad is found in BmVasa, DmVasa, Ddx4, and other DEAD-box proteins such as UAP56 and eIF4A (Supplementary Fig. 1d). eIF4A is required for translation initiation and UAP56 functions in the nuclear export of RNAs, including piRNA precursors[23,37]. In this study, we

substituted Arg470 within the BmVasa QRER tetrad (467–470) to alanine, producing the BmVasa R470A mutant. As expected, this mutant exhibited only marginal RNA-binding activity in cross-linking immunoprecipitation (CLIP) assays in vivo (Fig. 1g). Immunofluorescence showed that this mutant did not localize to or assemble Vasa bodies in BmN4 cells (Fig. 1h and Supplementary Fig. 1b). This indicates that the RNA-binding activity possessed by the central RNA helicase domain

**Fig. 1 | Both N-IDR and RNA helicase domain of BmVasa are required for Vasa body assembly and Ago3-piRISC biogenesis in BmN4 cells. a** Top: the domain structures of BmVasa WT, ΔN, and ΔC. Bottom: Immunoprecipitation/ western blotting shows that Flag-BmVasa WT and the ΔC mutant, but not ΔN mutant, associate with Myc-BmVasa. Flag-EGFP: a negative control. Anti-Flag antibody was used for immunoprecipitation and western blotting (lower) and Anti-Myc antibody for western blotting (upper). **b, c** Subcellular localization of Flag-BmVasa WT (Flag-WT), Flag-BmVasa ΔN (Flag-ΔN), and Flag-BmVasa ΔC (Flag-ΔC) in BmN4 cells (Control) (**b**) and in BmN4 cells lacking endogenous BmVasa (BmVasa KD) (**c**) (green). DAPI (blue): nuclei. Scale bar: 5 μm. Right: enlarged images of the insets. **d** piRNAs loaded onto Flag-Ago3 in BmVasa-depleted BmN4 cells (BmVasa KD). Myc-BmVasa WT, ΔN, and ΔC were expressed prior to immunoprecipitation. Myc-EGFP: a negative control. Flag-IP (upper): $^{32}$P-labeled piRNAs bound to Flag-Ago3. Flag-IP (lower): Flag-Ago3 in the immunoprecipitates. Input: Flag-Ago3 (top) and Myc-BmVasa WT, ΔC, ΔN, and EGFP (middle). β-Tubulin (β-Tub) (bottom): a loading control. **e** Subcellular localization of endogenous Ago3 in cells in (**c**). Scale bar: 5 μm. Right: enlarged images of the insets. **f** Subcellular localization of Flag-BmVasa N-EGFP-C mutant (Flag-N-EGFP-C) in BmN4 cells (Control; left) and cells lacking endogenous BmVasa (BmVasa KD; right) (green). DAPI (blue): nuclei. Scale bar: 5 μm. Right: enlarged images of the insets. The domain structure of BmVasa N-EGFP-C is shown on top. **g** CLIP shows that WT BmVasa binds to RNAs in BmN4 cells but the R470A mutant does not. Western blotting (WB) shows that the amounts of WT BmVasa and its R470A mutant in the samples are approximately equal. **h** Subcellular localization of Flag-R470A in BmN4 cells (Control; left) and cells lacking endogenous BmVasa (BmVasa KD; right) (green). DAPI (blue): nuclei. Scale bar: 5 μm. Right: enlarged images of the insets. The domain structure of BmVasa R470A mutant is shown on top. X: Arg470 mutagenized to alanine. **i** CLIP showed that WT BmVasa and ΔC, but not ΔN, bind to RNAs in vivo. Western blotting (WB) shows that the amounts of WT BmVasa and its mutants in the samples are approximately equal. Source data are provided as a Source Data file.

through the QRER tetrad is required for BmVasa to localize to and assemble Vasa bodies. This is consistent with the findings for DmVasa, whose equivalent mutant R527A failed to localize to nuage in the ovaries[38]. It is noted that the R470A mutant often appeared to be peri-nucleolar when endogenous BmVasa was absent, although its physiological significance remains unknown. The R470A mutant of BmVasa contains the whole of N-IDR and, as expected, self-associated similarly to the WT (Supplementary Fig. 1e). These results suggest that the self-association activity of BmVasa via N-IDR is independent of BmVasa–RNA binding.

CLIP assays showed that the ΔC mutant binds RNAs in vivo (Fig. 1i). In sharp contrast, no such activity was found for the ΔN mutant, although both mutants contained the RNA helicase domain intact (Fig. 1i). BmVasa N-IDR alone showed very low RNA-binding capacity in vitro: in fact, it was much lower than that of WT BmVasa (Supplementary Fig. 1f). These results, together with those in Fig. 1g, indicate that N-IDR may have minimal affinity for RNA but is clearly insufficient to confer full RNA-binding activity of BmVasa. Rather, N-IDR supports the central RNA helicase domain to confer full RNA-binding activity of BmVasa via self-association.

The expression of ΔN mutant instead of endogenous BmVasa in BmN4 cells produced very little Ago3-piRISC (Fig. 1d). Similarly, the R470A mutant of BmVasa was unable to produce Ago3-piRISC (Supplementary Fig. 1g). Thus, Vasa body assembly and Ago3-piRISC biogenesis that takes place in Vasa bodies require both BmVasa–BmVasa and BmVasa–RNA interactions via the N-IDR and the RNA helicase domain, respectively. The functional characteristics of each BmVasa mutant analyzed so far in this study are summarized in Supplementary Fig. 1h.

#### Recombinant BmVasa alone can assemble nuage-like droplets via phase separation in vitro

To track the dynamics of BmVasa in living BmN4 cells, EGFP-BmVasa was expressed in BmN4 cells and live imaging was performed under a microscope. Vasa bodies emitting green fluorescence were detected and observed to fuse with one another over time, a typical feature of LLPS-driven structures (Fig. 2a, b and Supplementary Movie 1). When the cells were treated with 200 mM ammonium acetate, which disturbs electrostatic intermolecular interactions, the fluorescent bodies mostly disappeared (Fig. 2c). However, when the cells were replaced with fresh medium without the salt, the fluorophores began to reappear (Fig. 2c). This liquidus, spontaneous coalescence and dispersion of the bodies strongly suggest that Vasa bodies in BmN4 cells are driven by LLPS.

To examine the ability of BmVasa alone to undergo LLPS, fluorescently labeled recombinant BmVasa (Supplementary Fig. 2a) was incubated in 96-well culture plates in the presence of 15% Polyethylene Glycol (PEG) 6000, a widely used macromolecular crowding reagent. Spherical fluorescent droplets were observed microscopically

(Fig. 2d). We observed that these droplets fused over time (Fig. 2e) similar to Vasa bodies in BmN4 cells (Fig. 2a). The in vitro droplets were hardly observed in the presence of 500 mM ammonium acetate (Fig. 2f). We thus assert that BmVasa has the intrinsic ability to undergo LLPS on its own and that this ability is better exerted in the presence of support such as PEG 6000.

Recombinant BmVasa used in the in vitro assays had no symmetrical dimethyl arginine (sDMA) because it was produced in *E. coli*. However, a previous study showed that the substitution of arginine residues to lysine at N-IDR (RK mutant) of BmVasa, which is defective in sDMA modification, resulted in delocalization to nuage in vivo[27]. In this study, we conducted the in vivo experiments and found that the RK mutant localized to nuage (Supplementary Fig. 2b). The fluorescence recovery after photobleaching (FRAP) assay in vivo showed that the mutant has similar liquidity to WT (Supplementary Fig. 2c). Furthermore, this mutant was able to assemble Vasa bodies de novo in cells lacking endogenous BmVasa (Supplementary Fig. 2b). Taking these findings together, we state that the Vasa body formation in BmN4 cells is independent of sDMA modification of BmVasa N-IDR. Prmt5 is responsible for the sDMA modification of DmVasa and it was reported that, in *prmt5* mutant flies, the localization of DmVasa in ovarian germ cells did not change[39]. This is consistent with our findings in silkworm BmN4 cells.

Although sDMA modification of the N-IDR of BmVasa is not essential for Vasa body formation, the N-IDR was indispensable for Vasa body formation in vivo (Fig. 1c). We then assessed whether this was also true in vitro. To this end, recombinant BmVasa ΔN mutant was prepared (Supplementary Fig. 2a), labeled with ATTO488 (green fluorescent dye), and incubated in 96-well plates containing 15% PEG 6000. The ΔN mutant showed difficulty assembling droplets (Fig. 2g and Supplementary Fig. 2d, e), indicating that even in vitro BmVasa requires N-IDR-mediated self-association for nuage-like droplet formation. In the following assays, recombinant BmVasa R470A mutant was employed (Supplementary Fig. 2a). This mutant assembled the fluorescent droplets similarly to the WT control (Fig. 2g and Supplementary Fig. 2d), which suggests that, at least in vitro, BmVasa can induce phase separation independently of its RNA-binding activity in the presence of 15% PEG 6000, although in vivo BmVasa required both BmVasa–BmVasa and BmVasa–RNA interaction to induce LLPS-driven nuage formation.

#### The ability of BmVasa to assemble nuage-like droplets in vitro is enhanced by BmVasa–RNA interaction

We inferred that the discrepancy between the in vivo and in vitro assays might be attributable to the relatively high concentration of PEG 6000 (15%), although it has commonly been used in similar studies[40,41]. Therefore, we next conducted the assays with a reduced concentration of PEG 6000 (i.e., 5%). Under these conditions, WT BmVasa showed difficulty assembling fluorescent droplets (Fig. 3a and Supplementary

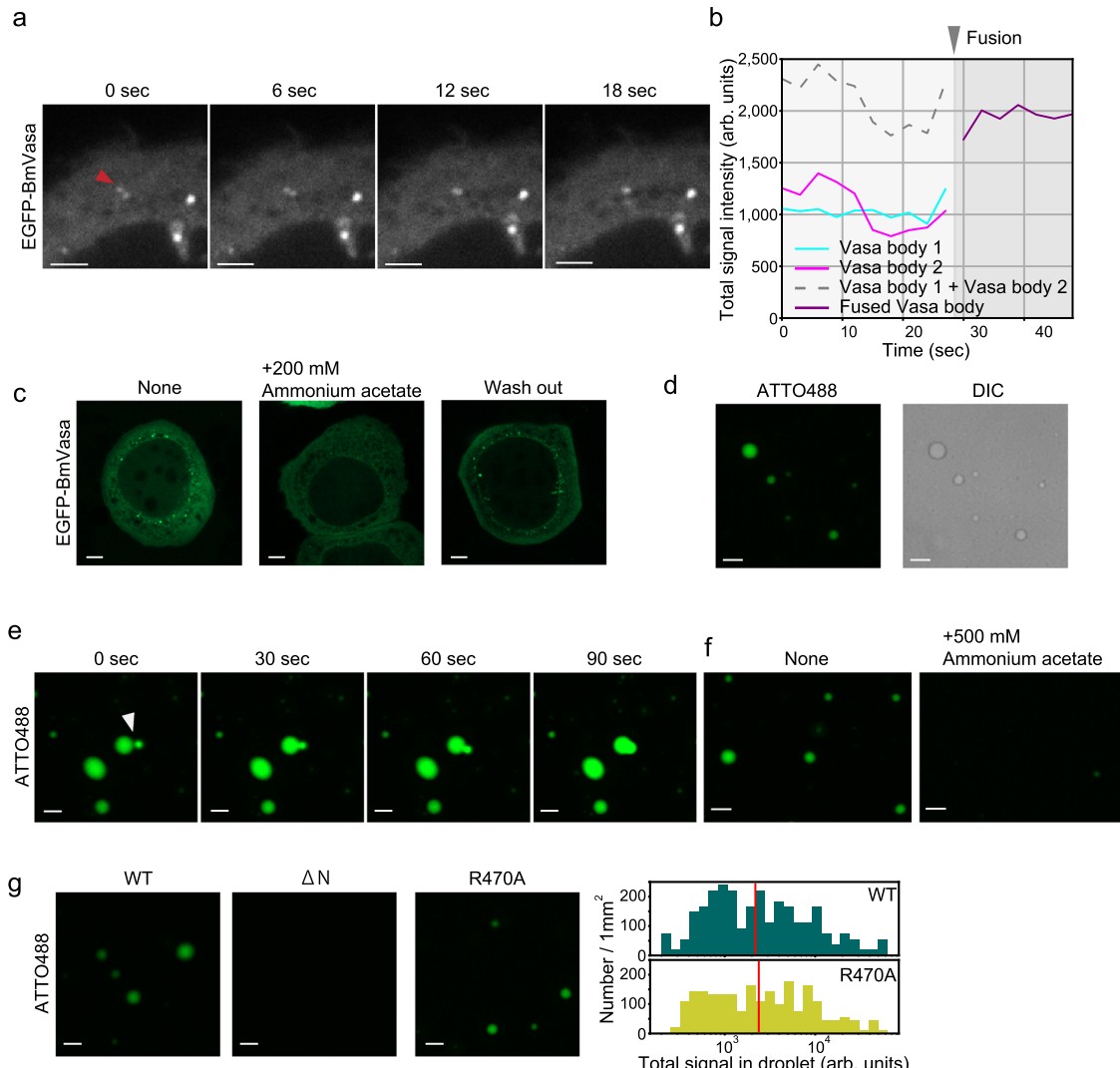

**Fig. 2 | Recombinant BmVasa has the ability to assemble nuage-like droplets in vitro. a** Snapshots of live imaging of EGFP-BmVasa in BmN4 cells. Two cytoplasmic EGFP-BmVasa-positive droplets indicated by a red triangle merge over time. Scale bar: 2 μm. **b** Signal intensity of Vasa body before and after fusion was quantified. **c** EGFP-BmVasa-positive droplets in BmN4 cells (None) disappeared upon treatment with 200 mM ammonium acetate. After washing out the salt, BmVasa-positive droplets were re-formed. Scale bar: 5 μm. **d** In vitro droplet formation assays in the presence of 15% PEG 6000 showed that 1.5 μM ATTO488-labeled recombinant BmVasa has activity to assemble nuage-like droplets. DIC: Differential interference contrast. Scale bar: 5 μm. **e** Snapshots of live imaging of 1.5 μM ATTO488-labeled recombinant BmVasa show that the BmVasa-positive droplets merge over time. Scale bar: 5 μm. **f** The BmVasa-positive in vitro droplets are not assembled in the presence of 500 mM ammonium acetate (BmVasa: 1.5 μM). Scale bar: 5 μm. **g** Assembly of BmVasa-positive droplets in vitro depends on self-association activity of BmVasa but not on its RNA-binding activity (BmVasa: 1.5 μM). Scale bar: 5 μm. Histograms of total ATTO488 signal intensity of a droplet are shown. Red line indicates the median [$p = 0.658$ (WT vs. R470A), by two-tailed Mann–Whitney U test]. Source data are provided as a Source Data file.

Fig. 3a). However, upon the addition of 2000 nt ssRNAs to the reaction mixture, droplets appeared (Fig. 3a and Supplementary Fig. 3a). When 50 nt ssRNAs were added instead, even though the total mass of RNAs added was adjusted to be about the same, few droplets appeared (Fig. 3a and Supplementary Fig. 3a, b). The presence of double-stranded RNAs (50 nt) also did not induce droplet formation (Supplementary Fig. 3b). These findings indicate that the BmVasa–ssRNA association induces BmVasa to undergo LLPS in vitro but its effectiveness depends RNA length. The low influence of RNA secondary structure was suggested from the observation that two 2000 nt ssRNAs with different sequences induced droplet formation to similar extents (Supplementary Fig. 3b). In contrast, the R470A mutant of BmVasa, which lacks RNA-binding activity, showed difficulty assembling fluorescent droplets even in the presence of 2000 nt ssRNAs (Fig. 3a and Supplementary Fig. 3a). The conditions with 5% PEG 6000 and 2000 nt ssRNAs appear to reflect the intracellular circumstances

more realistically than those with 15% PEG 6000 and an absence of ssRNAs.

The E339Q mutant of BmVasa remains bound to RNA and hydrolyzed ATP (i.e., ADP and inorganic phosphate), so its conformation remains fixed in a closed form with the substrates[27]. Glu339 is the second residue in the DEAD box (338–341) (Supplementary Fig. 1a). To understand how such a unique property of the E339Q mutant affects in vitro droplet formation, we prepared recombinant BmVasa E339Q mutant (Supplementary Fig. 2a) and performed in vitro assays along with WT BmVasa as a reference. The E339Q mutant could assemble the droplets similarly to WT BmVasa in the presence of 2000 nt ssRNAs (Fig. 3b and Supplementary Fig. 3c). The total amounts of E339Q signal per droplet were similar (Fig. 3b). We then added ATP and Mg²⁺ ions to the reaction mixture to allow BmVasa to hydrolyze ATP. Under these conditions, WT BmVasa presumably continues its binding onto and dissociation from RNAs[26]. Both WT BmVasa and its E339Q mutant

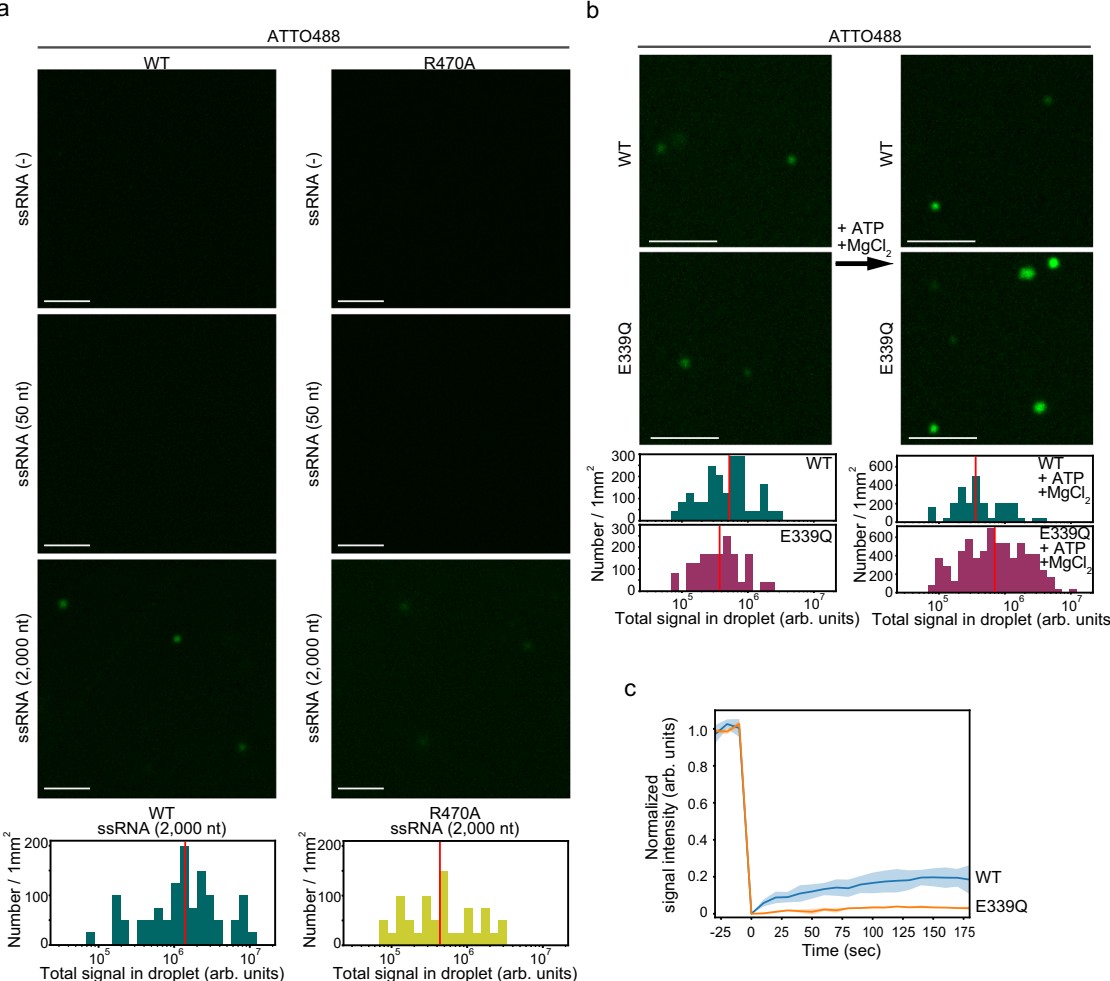

**Fig. 3 | The ability of recombinant BmVasa to assemble in vitro droplets is enhanced by BmVasa–RNA interaction. a** Neither ATTO488-labeled recombinant BmVasa WT nor R470A assembles the droplets in vitro in the presence of 5% PEG 6000. The addition of 2000 nt-long ssRNAs, but not shorter ones such as those of 50 nt-long ssRNAs, promotes the droplet assembly of BmVasa WT in vitro. BmVasa R470A mutant only weakly assembled the droplets even in the presence of 2000-nt-long ssRNAs. BmVasa: 1.5 μM. Scale bar: 5 μm. Histograms of total ATTO488 signal intensity of a droplet in the presence of 2000 nt-long ssRNAs are shown. Red line indicates the median [$p = 1.56 \times 10^{-4}$ (WT vs. R470A), by two-tailed

Mann–Whitney U test]. **b** Images of droplets of BmVasa WT and E339Q in the presence of 2000-nt-long ssRNAs before and after adding ATP and MgCl₂ (BmVasa: 1.5 μM). Scale bar: 5 μm. Histogram shows the total signal intensity within a droplet. Red line indicates the median [$p = 0.204$ (before adding ATP), $1.05 \times 10^{-3}$ (after adding ATP), by two-tailed Mann–Whitney U test]. **c** FRAP analysis of the droplet of ATTO488-labeled recombinant BmVasa WT and E339Q in the presence of 2000-nt-long ssRNAs, ATP, and MgCl₂. Mean (line) and standard deviation (shade) are shown ($n = 4$, individual data are plotted in Supplementary Fig. 3d). Source data are provided as a Source Data file.

formed droplets. However, the median E339Q signal overall was significantly stronger than the WT signal (x1.95) (Fig. 3b). This indicates that the droplet-forming capacity of the E399Q mutant is stronger than that of the WT. We found this result plausible because, as noted earlier, the E339Q mutant remains bound onto RNAs even in the presence of ATP and Mg²⁺ ions.

When the FRAP assay was performed on the droplets in vitro, WT BmVasa returned to the droplets more rapidly than the E339Q mutant did (Fig. 3c and Supplementary Fig. 3d). This indicates that WT BmVasa has higher dynamics than the E339Q mutant. It seems that such high dynamics in BmVasa significantly affects Vasa body turnover or BmVasa entry and exit into these bodies, but is not required for in vivo nuage assembly.

## Peptide composition of N-IDR influences the ability of Vasa to assemble nuage-like structures

The BmVasa mutant, N-EGFP-C, which is essentially equivalent to the Ddx4^YFP mutant, did not assemble speckles in either the nucleus or the cytoplasm (Fig. 1f), unlike the situation in Ddx4[33]. This suggests that

the N-IDRs of Ddx4 and BmVasa have distinct characteristics. The peptide sequence of the Ddx4 N-IDR showed that it contains 14 phenylalanines and the occupancy within the region is 5.9%, whereas the BmVasa N-IDR contains only 2 phenylalanines out of 135 amino acids (i.e., 1.5%), much less than that of the human counterpart (Fig. 4a, b). Phenylalanine contacts an arginine residue in the same protein molecules via cation-π interaction, a noncovalent interaction between the π-electron cloud of the aromatic ring and cation, facilitating the LLPS-driven assembly of protein condensates[42–44]. This may allow the N-IDR of Ddx4 to initiate the formation of nuage-like speckles without relying on RNA binding via the RNA helicase domain, unlike the N-IDR of BmVasa. There was no significant difference in the occupancy of arginine residues in the two N-IDRs, and the occupancy of the other residues was almost the same (Fig. 4b). Considering that negatively charged amino acids are also enriched in the N-IDR of BmVasa, charge blockiness in N-IDR[45] and RNA binding via the helicase domain cooperatively drive the multivalent interaction for LLPS of BmVasa.

To determine whether the difference in N-IDR characteristics between Ddx4 and BmVasa actually affects the assembly of nuage-like

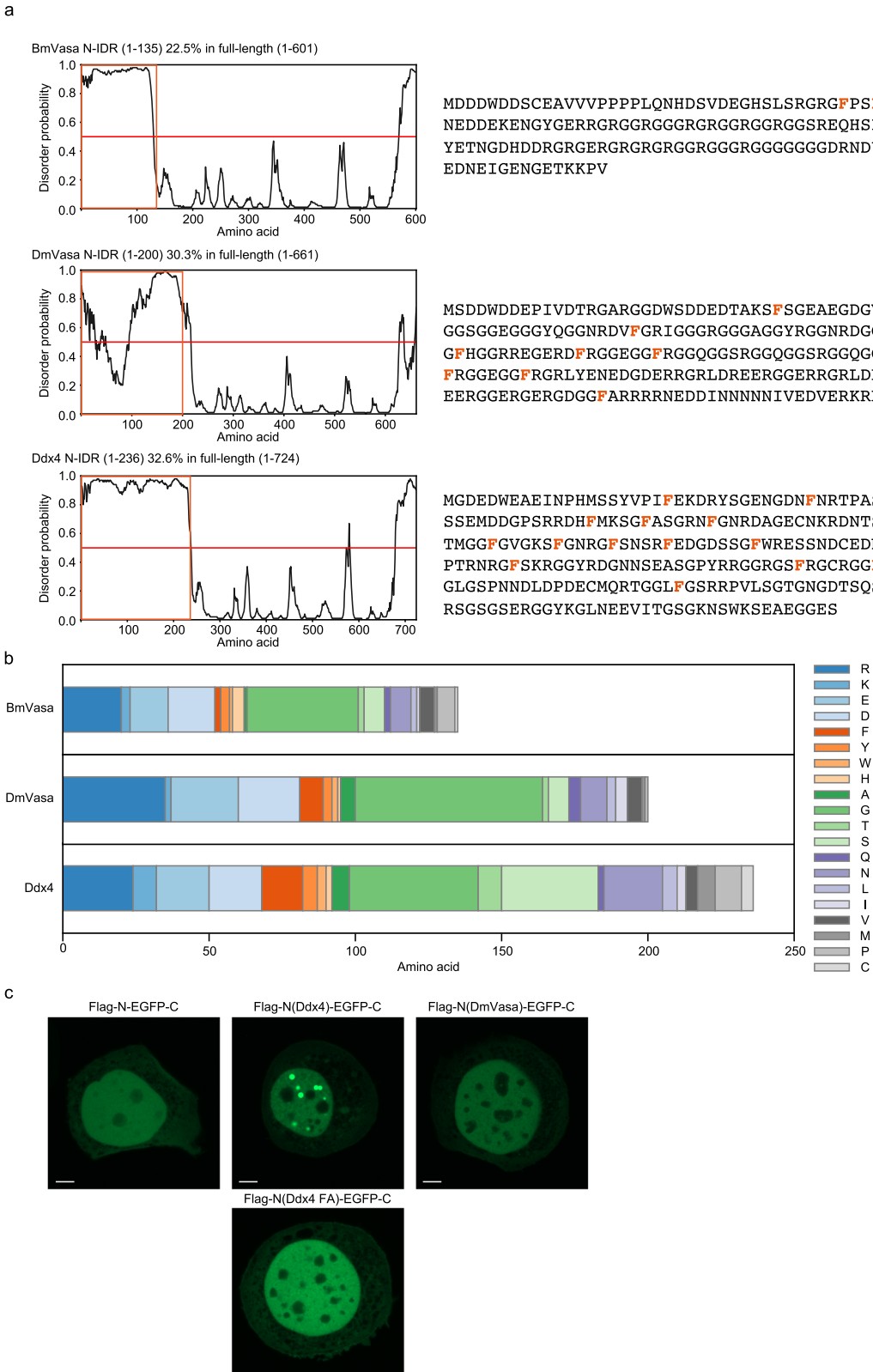

**Fig. 4 | Peptide composition of N-IDR affects the assembly of nuage-like structures. a** Disorder probability of BmVasa, DmVasa (N-terminal region of DmVasa corresponding to N-IDR of BmVasa), and Ddx4[33] (left). Orange open box and red line on the disorder plots represent N-IDR and disorder probability = 0.5, respectively. Peptide sequences of N-IDR of BmVasa, DmVasa and Ddx4 are also shown (right). Phenylalanine is indicated in orange. **b** Amino acid composition of N-IDR of BmVasa, DmVasa and Ddx4. **c** N-terminus of Flag-BmVasa N-EGFP-C was substituted with N-IDR of Ddx4 [Flag-N(Ddx4)-EGFP-C] and DmVasa [Flag-N(DmVasa)-EGFP-C]. All 14 phenylalanines in the N-terminus of Flag-N(Ddx4)-EGFP-C were substituted to alanine [Flag-N(Ddx4 FA)-EGFP-C]. Flag-BmVasa N-EGFP-C, Flag-N(Ddx4)-EGFP-C, Flag-N(Ddx4 FA)-EGFP-C, and Flag-N(DmVasa)-EGFP-C were expressed in BmN4 cells and their localization was observed based on the EGFP signal. Scale bar: 5 μm. Source data are provided as a Source Data file.

structures, the N-IDR of N-EGFP-C mutant was replaced with Ddx4 N-IDR and subcellular localization of the mutant in BmN4 cells was investigated fluorescently. Green fluorescent droplets appeared in the nuclei of BmN4 cells (Fig. 4c), as in HeLa nuclei expressing the Ddx4^YFP mutant. To test whether the presence of phenylalanine is the key factor determining this, all 14 phenylalanines in Ddx4 N-IDR of N(Ddx4)-EGFP-C were converted to alanine. The N(Ddx4 FA)-EGFP-C mutant could no longer assemble the nuclear structures (Fig. 4c). The importance of phenylalanine in the phase separation was thus demonstrated.

To gain insight into the interspecies relationship, we next decided to examine DmVasa N-IDR. DmVasa N-IDR is composed of 200 amino acids with phenylalanine constituting 4.0% of the residues (i.e., 8/200 residues) (Fig. 4a, b and Supplementary Fig. 4). From this information, it was inferred that DmVasa N-IDR may exhibit properties intermediate between those of Ddx4 and BmVasa N-IDRs, or may be similar to those of humans. However, the N(DmVasa)-EGFP-C mutant did not assemble nuage-like structures in BmN4 cells (Fig. 4c). There are several possible interpretations of this: for example, the length of DmVasa N-IDR is more similar to that of Ddx4, but DmVasa N-IDR has an intervening small valley with low disorder probability (Fig. 4a), which may make its properties more resemble those of BmVasa N-IDR.

## BmVasa preferably binds to transposon mRNAs in BmN4 cells

We next focused on which intracellular RNAs in BmN4 cells BmVasa binds to assemble nuage via phase separation. To address this, we performed FAST-iCLIP (Supplementary Fig. 5a). For comparison, total RNAs in the cells were sequenced genome-widely (Supplementary Fig. 5b). Because transposon derepression due to lack of Siwi-piRISC could affect BmVasa–RNA binding, both experiments were also performed in the absence of Siwi-piRISC (Supplementary Fig. 5a, b). To generate a total RNA library under such conditions, Siwi was depleted by RNAi. In FAST-iCLIP, Spindle-E (Spn-E) was depleted instead of Siwi because Siwi-depleted conditions did not yield a high-quality library. Previously, we showed that the loss of Spn-E reduces the level of Siwi-piRISC in BmN4 cells nearly as efficiently as the loss of Siwi[11].

First, total RNA sequence reads obtained under Siwi ± conditions were mapped to mRNA sequences of 14,300 protein-coding genes and 4296 transposons in the silkworm gene library (see "Methods"). Reads per kilobase of exon per million mapped reads (RPKM) were then calculated for each gene: the numbers of protein-coding genes and transposons with RPKM values greater than 10 were 2617 and 389, respectively. We interpreted these genes as those expressed in BmN4 cells and focused on these genes in subsequent analyses, unless otherwise noted.

To determine the populations of protein-coding genes and transposons expressed in normal BmN4 cells, we compared the RPKM values of the two groups. This analysis indicated that the populations of protein-coding genes and transposons were 73.3% (180,938 RPKM) and 26.7% (66,007 RPKM), respectively (Fig. 5a). A similar analysis was performed for BmN4 cells lacking Siwi using the total RNA reads. The populations for protein-coding and transposons were 65.5% (163,348 RPKM) and 34.5% (80,069 RPKM), respectively (Fig. 5b). The population of transposons increased significantly when Siwi was depleted. This was expected because transposon mRNAs are the primary target of Siwi-mediated silencing in BmN4 cells.

We next examined the FAST-iCLIP reads in the Siwi-piRISC ± libraries. The populations of BmVasa binding to protein-coding and transposon mRNAs were 40.3% [105,074 per million mapped reads (RPM)] and 59.7% (155,923 RPM) in normal cells and 31.6% (86,203 RPM) and 68.4% (186,896 RPM) in Siwi-piRISC-lacking cells, respectively (Fig. 5a, b). BmVasa-bound transposon mRNAs accounted for 59.7% in the control, but this increased to 68.4% in the absence of Siwi-piRISC. The rate of increase was lower than that of total RNA: namely,

BmVasa binding to transposon mRNA does not appear to be affected as much as transposon expression by Siwi-piRISC depletion. One hypothesis explaining this is that BmVasa may target and bind to RNAs that have just been exported from the nucleus and actively accumulate them in Vasa bodies. This idea is consistent with the observation that Vasa bodies tend to reside around the nucleus. Otherwise, transposon mRNAs would bind to the ribosome and become subject to translation, and silencing would not work.

## BmVasa binds to transposon mRNA with little dependence on its abundance, unlike for protein-coding mRNA

We then analyzed the correlation between total RNA reads (i.e., mRNA quantity) and FAST-iCLIP reads (i.e., BmVasa binding) in normal cells. Spearman's correlation coefficient for protein-coding genes was 0.378 (Fig. 5c), while for transposons it was 0.159 (Fig. 5c). These results suggest that BmVasa tends to bind to protein-coding mRNAs according to their abundance, but less so for transposon mRNAs.

To examine BmVasa binding to transposon mRNAs further, 389 transposons were classified by the degree of derepression due to the lack of Siwi-piRISC and analyzed (Fig. 5d). In the plots, the red dots indicate transposons whose expression was elevated relatively highly in the absence of Siwi-piRISC ($\log_2$ FC > 2). The green dots are those whose expression increased moderately in the absence of Siwi-piRISC ($1 < \log_2$ FC $\leq$ 2), and the transposons in blue were largely unaffected ($\log_2$ FC $\leq$ 1). In the BmVasa binding panel, all dots appeared to be distributed along the diagonal line (Fig. 5d). This agrees with our earlier notion that BmVasa binds to transposon mRNAs with little effect on changes in abundance due to Siwi-piRISC depletion.

## BmVasa binds to transposon mRNA with little dependence on its length, unlike for protein-coding mRNA

To determine the effect of RNA length on BmVasa–RNA binding, both protein-coding and transposon mRNAs were binned according to their lengths, and the expression and BmVasa-RNA binding levels (RPKM) were examined for each bin. In protein-coding mRNAs, both total (gray) and BmVasa-RNA binding (yellow) peaked around 1000 nt (Fig. 5e). Their distribution patterns appeared to be similar. However, calculation of enrichment (BmVasa binding/Total) (red) showed a tendency for BmVasa to preferentially bind to relatively long mRNAs, around 4500 nt or possibly longer. Such long RNAs might render Vasa bodies more stable, and this idea is not in conflict with the results of in vitro analysis. However, no such clear trend was observed in transposon mRNAs (Fig. 5e), suggesting that BmVasa binds to them in a manner less dependent on their length than for protein-coding mRNAs.

We then extracted transposons annotated as LINE, SINE, LTR, and DNA (see "Methods"), and examined the BmVasa-RNA binding status for each category. SINE transposons are rather short in general, and this was indeed the case for silkworm (Supplementary Fig. 5c). However, the examination showed that BmVasa bound to SINE mRNAs at a level similar to those for other types of transposon mRNAs (Supplementary Fig. 5c). Thus, although BmVasa prefers transposon mRNAs to bind to and assembles Vasa bodies in germ cells, its preference is not simply based on length.

If BmVasa–RNA binding were determined simply by the primary sequence of RNA, such differences in dependence on RNA length and/or abundance would not have been observed between transposon and protein-coding mRNAs. In fact, our sequence analysis of FAST-iCLIP data did not yield a definitive conclusion. It is inferred that the silkworm germ cells have sophisticated and selective machinery that actively promotes selective binding of BmVasa and transposon mRNAs. However, its machinery is not strict enough to exclude protein-coding mRNAs. BmVasa also binds to them weakly, but it largely depends on their abundance and length. This may also apply to certain types of transposon mRNAs.

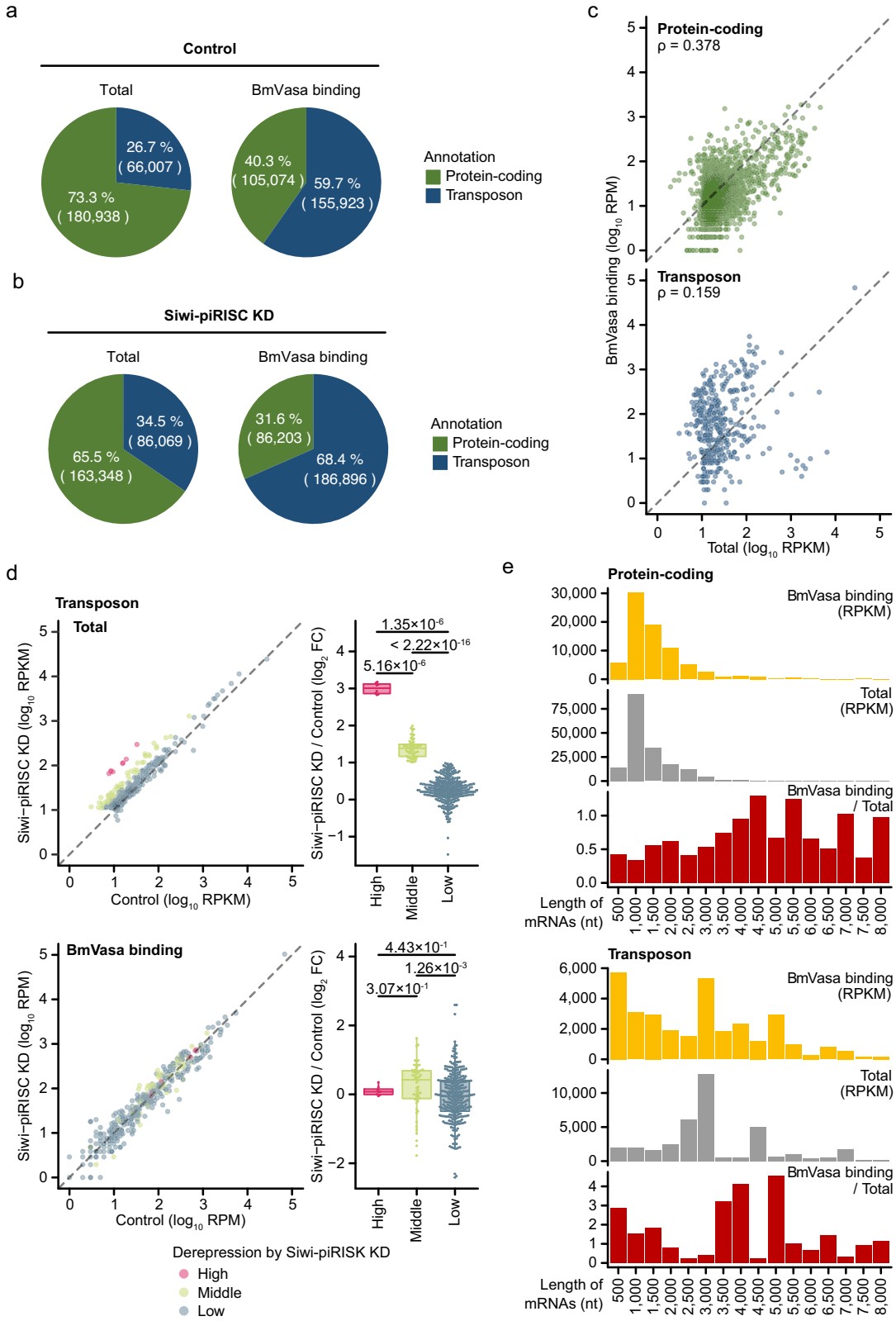

## Discussion

Vasa is widely used in animals as a germ cell marker, and its molecular function has been studied for many years. One such study revealed that the RNA helicase domain of DmVasa alone exhibits sufficient RNA-binding activity[26]. However, the present study focusing on BmVasa demonstrated that the RNA-binding activity of the RNA helicase domain alone is marginal, but is greatly enhanced by N-IDR, the self-

association domain of BmVasa, whose RNA-binding capacity is also essentially negligible. This suggests that the dependence of Vasa on the RNA helicase domain and N-IDR for exhibiting its function differs among species. In fact, the N-IDR exchange experiments conducted in this study further highlighted this aspect: human Vasa (Ddx4) N-IDR formed nuage-like droplets by itself, while BmVasa and DmVasa N-IDRs showed no such ability. Contrary to these findings, the vital role of Vasa

**Fig. 5 | BmVasa preferentially binds to transposon mRNAs in vivo. a, b** Pie charts show the proportions of transposon mRNAs (blue) and protein-coding mRNAs (green) in the total RNA sequencing library (Total; left) and in the BmVasa FAST-iCLIP library (BmVasa binding; right), both of which were individually produced in BmN4 cells (Control) (**a**) and in BmN4 cells lacking Siwi-piRISC (Siwi-piRISC KD) (**b**). The association between the data is statistically significant (X $^2$ = 56,205, df = 1, $p$ value <2.2 × 10$^{-16}$, by Pearson's chi-squared test (Two-tailed)) (**a**), (X $^2$ = 60,140, df = 1, $p$ value <2.2 × 10$^{-16}$, by Pearson's chi-squared test (Two-tailed)) (**b**). **c** Scatter plots show correlation between RPKM values of the total RNA sequencing library (Total) and RPM values of the BmVasa FAST-iCLIP library (BmVasa binding) for protein-coding mRNAs (green; top) or transposon mRNAs (blue; bottom). ρ is Spearman's rank correlation coefficient. **d** Scatter plots (left) show RPKM values of the total RNA sequencing library (Total; top) or RPM values of the BmVasa FAST-

iCLIP library (BmVasa binding; bottom) for transposon mRNAs. They were categorized by the FCs in the expression levels of the mRNAs produced in BmN4 cells lacking Siwi-piRISC (Siwi-piRISC KD) relative to controls into "High" (red, log$_2$ FC > 2, $n$ = 8), "Middle" (green, 1 < log$_2$ FC ≤2, $n$ = 58), or "Low" (blue, log$_2$ FC ≤1, $n$ = 323). Box plots (right) show the FCs as in the scatter plots. Two-tailed Mann–Whitney U test was performed and p values are indicated on the box plots. Center line: median. Box limits: upper and lower quartiles. Whiskers: 1.5× interquartile range. **e** Bar graphs show the length distribution (8,000 nt or less) of protein-coding mRNAs (upper 3 graphs) and transposon mRNAs (lower 3 graphs). RPKM values of the BmVasa FAST-iCLIP library (BmVasa binding; top), RPKM values of the total RNA sequencing library (Total; middle), and the enrichment value (BmVasa binding relative to Total; bottom) are respectively indicated. Source data are provided as a Source Data file.

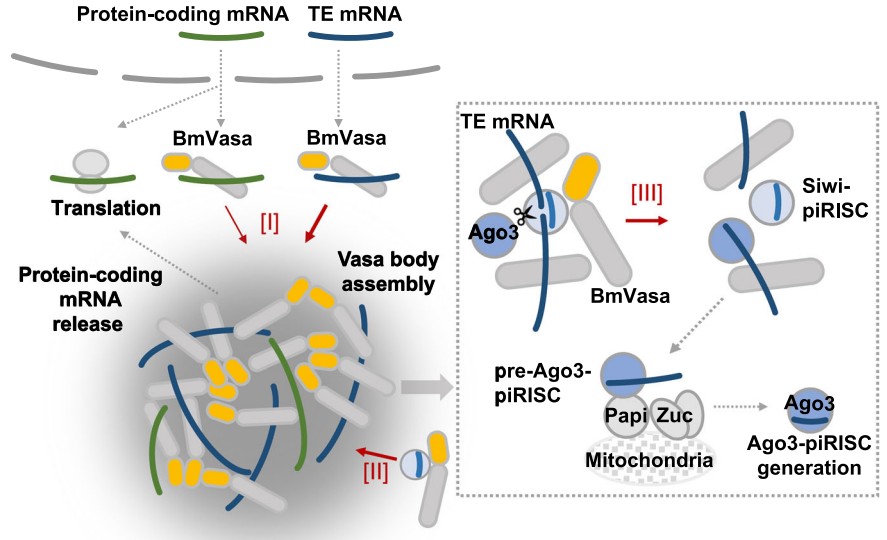

**Fig. 6 | The model of BmVasa functions in Ago3-piRISC biogenesis.** Functions of BmVasa in Ago3-piRISC biogenesis: [I] Vasa body assembly upon BmVasa–RNA binding. BmVasa binds to transposon mRNAs more preferentially than to protein-coding mRNAs. Transposon mRNAs in Vasa bodies are consumed by Siwi-piRISC to produce Ago3-piRISC. Protein-coding mRNAs are released from Vasa bodies by the ability of BmVasa to bind to and dissociate from bound RNAs upon ATP hydrolysis. This reaction may also occur for transposon mRNAs, but the quicker the transposon mRNAs are cleaved by Siwi-piRISC, the higher the silencing effect is expected to be. [II] BmVasa binds to Siwi-piRISC and brings it to Vasa bodies. [III] Cleaved RNAs

are actively released from Siwi-piRISC by BmVasa. This reaction also relies on the ability of BmVasa to dissociate from bound RNAs upon ATP hydrolysis. Released RNA is bound to Ago3, giving rise to Ago3-piRISC precursor (pre-Ago3-piRISC). The pre-Ago3-piRISC then moves to the mitochondrial surface, where it is further processed in cooperation with Zuc and Papi[30]. Yellow and gray ovals: N-IDR and RNA helicase domain of BmVasa, respectively. BmVasa C-IDR is omitted for simplification. N-IDR of BmVasa binding to transposon mRNA is omitted for simplification in the dotted box.

function in the piRNA pathway is conserved among these three organisms. This study clearly contrasted the interspecies conservation of Vasa "protein" function with diversity of Vasa's "domain" functions, adding to the value of this work. The piRNA pathway itself is highly conserved among animal species, but once we focus on the factors required for the pathway and their functional contributions to it, we find that they are less conserved across species than we would expect. It appears as if each organism readily arranges its piRNA mechanism according to the environment in which it is located. piRNAs and transposons are in a constant arms race. This rapid adaptability may have been necessary for the piRNA pathway of each species to respond the transposons of each species. Something similar has been discussed before[46], and the findings of this study further support this statement.

BmVasa preferentially binds to transposon mRNAs and assembles Vasa bodies, the centers of Ago3-piRISC biogenesis, although our current analysis did not allow us to identify mechanisms controlling the selectivity of BmVasa–transposon mRNA. BmVasa appeared to

bind not only transposon but also protein-coding mRNAs, although differences were evident in the degree of preference. Transposon mRNAs sequestered into Vasa bodies via BmVasa are ultimately destined to be cleaved by Siwi-piRISC. The cleaved RNAs are then processed to piRNAs for Ago3 (Fig. 6). Meanwhile, very few Siwi-piRISCs consume protein-coding mRNAs in Vasa bodies. This is simply because piRNAs against protein-coding genes essentially constitute a very minor component. Based on this, one would imagine that protein-coding mRNAs accumulate in Vasa bodies for suppression, like in P-bodies or stress granules under certain intracellular conditions. However, the molecular characteristics of Vasa (not limited to silkworm but also applying to other species), that is, hydrolyzing ATP and using the energy to repeatedly bind and dissociate RNAs, suggest that this is unlikely to be the case. This reaction can be interpreted as being sufficient to eject protein-coding mRNAs or any other RNAs not targeted by Siwi-piRISC, including leftovers of piRNA processing, from Vasa bodies. This idea is supported by the earlier observation that

expression of the E339Q mutant of BmVasa, defective in ATP hydrolysis and RNA release, instead of endogenous BmVasa in BmN4 cells resulted in the abnormal enlargement of Vasa bodies[27]. This study corroborated that this is also true in vitro.

Silkworms do not have a homolog of Piwi. This means that, unlike mice and flies, they have no piRNAs repressing transposons at the transcriptional level. In other words, piRNA-targeted transposons in BmN4 cells are presumably fully transcribed at a significant rate, as are other genes important for cellular life. This was in fact evident from the total RNA sequencing data in the absence of Siwi. Under these circumstances, BmVasa has a high probability of binding transposon mRNAs immediately upon their nuclear export. In the germ cells of mice and flies, transposons are repressed transcriptionally by Miwi2 and Piwi, respectively[47,48]. Therefore, the amounts of transposon mRNAs exported to the cytoplasm must be inherently limited in these two animals. To overcome this limitation, DmVasa may rely on nuclear UAP56, which selectively binds piRNA precursors in the nucleus and forwards them to cytoplasmic DmVasa. Because the depletion of UAP56 in BmN4 cells resulted in severe cell damage, we currently have no way of testing the possibility that UAP56 functions in piRISC biogenesis in BmN4 cells. Nxf3 in *Drosophila* appears to be specialized in the mechanism of piRNA (piRISC) biogenesis, where Rhino, a member of heterochromatin protein 1 family, plays an important role[24]. *Bombyx* has no genes homologous to *Rhino*. In our analysis, silkworm does not appear to have a homolog of *Nxf3* either. Thus, Nxf3 may not be the factor we are looking for. BmVasa binds preferentially to transposon mRNAs. It seems unlikely that BmVasa itself has a particular ability to accomplish such selectivity, so factors that bind to BmVasa may aid in the selectivity. Studies of these issues are underway in the laboratory.

# Methods

## Cell culture
BmN4 cells were cultured at 26 °C in EX-CELL 420 serum-free medium for insect cells (Sigma-Aldrich) supplemented with 10% fetal bovine serum (Gibco) and 1x penicillin-streptomycin-glutamine (Thermo Fisher Scientific).

## Plasmid construction
Vectors to express 3xFlag-EGFP, 3xFlag-BmVasa (WT and E339Q) and 3xFlag-Ago3 in BmN4 cells were constructed[11]. To yield pET47b-BmVasa WT and ΔN, *BmVasa* cDNA was amplified by PCR using pIB-3xFlag-BmVasa as a template and inserted into pET-47b using NEBuilder HiFi DNA Assembly Master Mix (New England Biolabs). pET47b-BmVasa R470A, E339Q, and N-IDR were prepared by inverse PCR using PrimeSTAR Max DNA polymerase (Takara). The expression vectors in BmN4 cells were constructed using pIB-V5-His-DEST vector. The expression vector of BmVasa R470A was prepared by inverse PCR using PrimeSTAR Max DNA polymerase. The expression vector of EGFP-BmVasa WT and BmVasa mutants [ΔN (136–601),ΔC (1–564), N-EGFP-C (1–135/EGFP/565–601), RK[27], N(Ddx4)-EGFP-C, EGFP-BmVasa RK, N(DmVasa)-EGFP-C, N(Ddx4 FA)-EGFP-C] was prepared by inverse PCR and fragment insertion by NEBuilder HiFi DNA Assembly Master Mix. pHR-DDX4N-mCh-Cry2WT[49], which was used as a PCR template of N-IDR of Ddx4, was a gift from Clifford Brangwynne (Addgene plasmid # 101225; http://n2t.net/addgene:101225; RRID: Addgene_101225). N(DmVasa) was amplified from pAcF-DmVasa. N(Ddx4 FA) was synthesized at Thermo Fisher Scientific. The primers used are summarized in Supplementary Table 1.

## IDR prediction and sequence alignment
The disorder profile was mapped via DISOPRED3[50]. The sequence alignment was performed with EMBOSS Needle[51].

## RNAi and transgene expression in BmN4 cells
BmN4 cells (0.5–1.5 × 10^6 cells) suspended in 100 μL of EP buffer [137 mM sodium chloride, 5 mM potassium chloride, 0.5 mM sodium

hydrogen phosphate, and 2.1 mM HEPES-KOH (pH 7.1)] were transfected with 500–750 pmol siRNA duplex by electroporation with Nucleofector 2b (Lonza). Luciferase (Luc) siRNA was used as a control. To exogenously express proteins in BmN4 cells, the cells were transfected with 2 μg of plasmid in 5 μL of FuGENE HD transfection reagent (Promega). After transfection, cells were incubated at 26 °C for 24–48 h. Sequences of the siRNAs used are summarized in Supplementary Table 1.

## Antibodies
The primary antibodies used in this study were anti-Flag M2 monoclonal antibody (1:5,000 dilution for western blotting and 1:1,000 dilution for immunofluorescence; Sigma), anti-Myc monoclonal antibody (1:5,000 dilution, 9E10; Developmental Studies Hybridoma Bank), anti-BmVasa antibody [1:20 dilution (supernatant), 1C3D10[11]], anti-Ago3 antibody (1:500 dilution for immunofluorescence, 7A7[11]), and anti-β-Tubulin monoclonal antibody (1:1,000 dilution, E7; Developmental Studies Hybridoma Bank). The secondary antibodies used in this study were Peroxidase-conjugated anti-mouse IgG (1:5,000 dilution; Cappel), TrueBlot ULTRA: Anti-Ig HRP, Mouse (Rat), eB144 (1:1,000 dilution; ROCKLAND), Goat anti-Mouse IgG1 Cross-Adsorbed Secondary Antibody, Alexa Fluor™ 488 (1:1,000 dilution; Invitrogen), and Goat anti-Mouse IgG2a Cross-Adsorbed Secondary Antibody, Alexa Fluor™ 555 (1:1,000 dilution; Invitrogen).

## Immunoprecipitation
For co-immunoprecipitation, BmN4 lysates prepared using Co-IP buffer [30 mM HEPES-KOH (pH 7.3), 150 mM potassium acetate, 5 mM magnesium acetate, 5 mM dithiothreitol, 0.1% (w/v) NP-40, 2 μg/mL pepstatin, 2 μg/mL leupeptin, and 0.5% aprotinin] were incubated with anti-Flag M2 monoclonal antibody (Sigma)-bound Dynabeads Protein G (Thermo Fisher Scientific) at 4 °C for 1 h. The beads were washed three times with Co-IP buffer and then isolated proteins were eluted with sodium dodecyl sulfate (SDS) sample buffer for western blotting.

## Western blotting
Proteins on SDS polyacrylamide gels were blotted on nitrocellulose membranes or polyvinylidene difluoride membranes. The membranes were blocked with 5% skimmed milk in phosphate-buffered saline (PBS; FUJIFILM Wako Pure Chemical) followed by incubation with primary antibodies diluted in 0.1% Tween-20 in PBS (T-PBS). After washing with T-PBS, membranes were incubated with secondary antibodies diluted in T-PBS. The membranes were then extensively washed with T-PBS and incubated with Clarity Western ECL Substrate (Bio-Rad). Imaging of chemiluminescence was performed using a Chemi-Doc XRS Plus System (Bio-Rad). To quantify the efficiency of self-association for BmVasa, Fiji[52] was used to measure signal intensities of the protein bands.

## Rescue assay
BmN4 cells lacking BmVasa were transfected with 2 μg of plasmids encoding Myc-tagged BmVasa WT and its mutants using FuGENE HD and incubated at 26 °C for 48 h. The cells were then transfected with 2 μg of Flag-Ago3 plasmid and incubated at 26 °C for 24 h. Immunoprecipitation against Flag-Ago3 was performed and analyzed by western blotting. Immunoprecipitated proteins were treated with proteinase K (Roche) for RNA extraction. RNAs were extracted using phenol−chloroform and precipitated with ethanol. RNA ³²P-labeling was carried out using T4 polynucleotide kinase (New England Biolabs).

## CLIP
BmN4 cells expressing Flag-EGFP or Flag-BmVasa variants were UV crosslinked with irradiance of 200 mJ/cm² at 254 nm UV. Flag-tagged proteins were immunoprecipitated as described above in the presence of RNase. The beads were incubated with Co-IP buffer containing

RNase at 26 °C for 15 min and washed three times with Co-IP buffer containing 500 mM NaCl. Dephosphorylation and $^{32}$P-labeling of crosslinked RNAs were performed using Antarctic Phosphatase and T4 polynucleotide kinase. Upon SDS polyacrylamide gel electrophoresis (SDS-PAGE), proteins were detected by western blotting. $^{32}$P-labeled RNAs were visualized using Typhoon FLA 9500 (Cytiva).

### Immunofluorescence
BmN4 cells were placed on cover glasses covered with 0.075% (w/v) poly-L-lysine. Cells were fixed with 4% formaldehyde in PBS for 15 min and permeabilized with 0.2% (v/v) Triton X-100 in PBS for 15 min. After blocking with 3% (w/v) BSA in PBS (PBS-B) for 30 min, cells were incubated with primary antibodies diluted with PBS-B for 1 h. After washing with PBS-B, cells were incubated with Alexa 488- or 555-labeled anti-mouse secondary antibodies in PBS-B for 1 h in the dark. Cover glasses were mounted with Vectashield Antifade Mounting Medium with DAPI (Vector Laboratories). For observation of EGFP signal, BmN4 cells on cover glasses were mounted just after fixation. Fluorescence imaging was performed with an LSM 980 confocal laser-scanning microscope (Zeiss) equipped with a Plan-Apochromat 63x/ 1.4 Oil DIC M27 objective lens (Zeiss). To quantify the expression level, Fiji[52] was used to obtain the mean signal intensity of cytoplasm in the BmN4 cells.

### Live cell imaging
BmN4 cells were placed on a 35 mm glass-bottomed dish (Matsunami) covered with 0.075% (w/v) poly-L-lysine. Fluorescence imaging was performed with an LSM 980 confocal laser-scanning microscope equipped with a Plan-Apochromat 63x/1.4 Oil DIC M27 objective lens at room temperature. For ammonium acetate treatment, the original culture medium was replaced by culture medium with 200 mM ammonium acetate. Then, cells were washed with PBS and fresh culture medium was added. For fluorescence recovery after photobleaching assay, the EGFP signal of a single Vasa body was bleached using light from a 488-nm laser and observed by time-lapse imaging. To quantify the fluorescence, Fiji[52] was used to obtain the integrated density of the bleached area.

### Purification of recombinant proteins
Rosetta2 (DE3) cells (Merck) were transformed with pET vectors encoding His-tagged proteins. The cells were grown at 37 °C in Luria Bertani medium until the optical density at 600 nm reached 0.6 and protein expression was induced by the addition of 0.2–0.5 mM iso-propyl β-D-1-thiogalactopyranoside (IPTG) at 16 °C overnight. To purify the His-tagged BmVasa variants, the cells were suspended with His A buffer [20 mM Tris-HCl (pH 8.0), 1 M NaCl, 20 mM imidazole, 1 mM dithiothreitol, and cOmplete ULTRA EDTA-free protease inhibitor (Roche)] and lysed by sonication. The samples were centrifuged and filtered to remove the cell debris, and then loaded onto Ni Sepharose 6 Fast Flow resin (GE Healthcare). The resin was washed with His A buffer and bound proteins were eluted with His B buffer [20 mM Tris-HCl (pH 8.0), 1 M NaCl, 300 mM imidazole, and 1 mM dithiothreitol]. The eluted proteins were dialyzed against dialysis buffer [50 mM Tris-HCl (pH 8.0) and 100 mM NaCl]. After dialysis, proteins were purified by anion-exchange chromatography and gel filtration using HiTrap SP FF (Cytiva) [to purify ΔN, EnrichQ (Bio-Rad) was used for cation-exchange chromatography] and ENrich SEC650 (Bio-Rad), respectively.

### In vitro transcription
DNAs with T7 promoter sequence were amplified from pAcM-lacZ (CDS 1–47 and 1–1997) and *Bombyx mori* protein-coding gene [BMgn003210: retrieved from the KAIKObase database[53]] (CDS 1–47 and 1–1997) using Q5 DNA polymerase (New England Biolabs). The DNA fragments were amplified again and size-selected by gel electro-phoresis. Amplified DNAs were extracted using phenol–chloroform

and precipitated with ethanol. In vitro transcription was performed with AmpliScribe T7 High Yield Transcription Kit (Lucigen). Transcribed ssRNAs were extracted using phenol–chloroform and precipitated with ethanol. The primers used are summarized in Supplementary Table 1. For dsRNA formation, antisense RNA of 50 nt lacZ was transcribed as described above. The same amounts of sense and antisense 50 nt lacZ RNA were mixed in annealing buffer [10 mM Tris-HCl (pH 8.0), 50 mM NaCl, and 1 mM EDTA]. The mixture was incubated at 95 °C for 3 min and gradually cooled to 25 °C.

### In vitro droplet formation assay
ATTO488-maleimide (ATTO-TEC) dissolved in DMSO (final 10 μM) was added to the protein solution and incubated in the dark for 60 min. DTT (final 5 mM) was then added to the solution and incubated for 60 min to quench the reaction. To observe the self-interaction in highly crowded reagent conditions, fluorescently labeled proteins were mixed with droplet buffer A [50 mM Tris-HCl (pH 8.0), 150 mM sodium chloride, and 15% (w/v) PEG 6000]. To check the effect of ammonium acetate, droplet buffer was supplemented with 500 mM ammonium acetate. To observe the RNA-dependent droplet formation, the labeled proteins were mixed with ssRNA and then diluted in buffer B [50 mM Tris-HCl (pH 8.0), 150 mM sodium chloride, and 5% (w/v) PEG 6000]. To study the effect of ATP, 4 mM ATP and 2 mM Mg$^{2+}$ were added to the mixture. The mixture was transferred to a 96-well plate and then observed with an LSM 980 confocal laser-scanning microscope equipped with a Plan-Apochromat 63x/1.4 Oil DIC M27 objective lens. For quantification of droplet formation with Fiji[52], several images taken from randomly chosen areas in the well were used. For droplet detection, the background was subtracted by "Subtract Background" (Rolling Ball Radius: 50 pixels, "Sliding paraboloid") and the images were binarized by "Threshold" ["Li" for droplets under 15% (w/v) PEG 6000 conditions and "Triangle" for droplets under 5% (w/v) PEG 6000 conditions]. Noise was reduced by "Binary > Open" and the region of interest (ROI) for each droplet was generated by "Analyze Particles." The ROI was applied to unprocessed original images and the integrated density of each ROI was obtained. Then, the frequency (the number/mm$^2$) of integrated density of each droplet was plotted as a histogram. For fluorescence recovery after photobleaching assay, ATTO488 signal in the whole droplet area was bleached using light from a 488-nm laser and observed by time-lapse imaging. To quantify the fluorescence, Fiji[52] was used to obtain the integrated density of the bleached area.

### Electrophoretic mobility shift assay
For the assay, 6.6 μM protein, 100 ng/μL of 50 nt lacZ ssRNA, and RNasin® Plus Ribonuclease Inhibitor (Promega) were mixed in EMSA buffer [50 mM Tris-HCl (pH 8.0) and 150 mM NaCl]. The solution was incubated on ice for 30 min. Then, the samples were subjected to nondenaturing polyacrylamide gel electrophoresis at 4 °C. The gel was stained with SYBR Gold. Imaging was performed using a ChemiDoc XRS Plus System (Bio-Rad). The image was quantified with "Plot Profile" of Fiji[52] along with the lane.

### Gene classification
To customize the gene annotation in this study, the model of 16,880 *Bombyx mori* genes (Jan. 2017) retrieved from the SilkBase database[54] and 1716 *Bombyx mori* transposons[31] were initially combined. Because the gene model contains transposon-like sequences, sequences with a description related to transposons in the gene information were classified as transposons. Besides, using the gene model as queries for BLASTN searches against the transposons, sequences whose coverage and identity were over 90% were additionally categorized as transposons. As a result, 18,596 sequences were classified as protein-coding genes (14,300 sequences) and transposons (4,296 sequences) and this pool was used as the silkworm gene library.

## FAST-iCLIP

FAST-iCLIP[55] basically uses the standard iCLIP protocol[56] except that pulldown is performed in the step of cDNA purification using a 3′-end biotin-blocked adaptor and streptavidin beads in place of size selection by gel electrophoresis. BmN4 cells in the two 100-mm dishes treated with Luc (control) or Spn-E siRNA were used. BmN4 cells were washed once with ice-cold PBS and UV crosslinking was performed with irradiance of 200 mJ/cm² at 254 nm. The cells were subjected to immunoprecipitation using anti-BmVasa antibody (1C3D10)[11]. The beads were washed three times with wash buffer [30 mM HEPES-KOH (pH 7.3), 150 mM potassium acetate, 5 mM magnesium acetate, 5 mM dithiothreitol, 0.1% (w/v) NP-40, 2 µg/mL pepstatin, 2 µg/mL leupeptin, and 0.5% aprotinin], followed by partial RNA digestion with 1 U/µL RNase T1 for 15 min at room temperature. Then, the beads were washed three times with high-salt wash buffer [30 mM HEPES-KOH (pH 7.3), 150 mM potassium acetate, 5 mM magnesium acetate, 500 mM NaCl, 5 mM dithiothreitol, 0.1% (w/v) NP-40, 2 µg/mL pepstatin, 2 µg/mL leupeptin, and 0.5% aprotinin]. 3′-end RNA dephosphorylation was performed with T4 polynucleotide kinase and the 3′-ends of the RNAs were ligated to a radioactively labeled and preadenylated RNA adaptor using T4 RNA ligase 2, truncated KQ (New England Biolabs). The samples were analyzed by NuPAGE (Life Technologies) and RNA–protein complexes were transferred to nitrocellulose membranes. The RNA–protein complexes on nitrocellulose membranes were treated with proteinase K. Extracted RNAs were subjected to reverse transcription using SuperScript III Reverse Transcriptase (Life Technologies), followed by RNA degradation using RNaseA (Roche) and RNaseH (NEB). cDNA purification and cDNA circularization were performed by pulldown using Pierce™ Streptavidin Magnetic Beads (Life Technologies) and CircLigase™ II ssDNA Ligase (Epicentre). Libraries were then amplified with Q5 DNA polymerase and size-selected by gel electrophoresis. The libraries were sequenced on the MiSeq platform (Illumina) to obtain 51 nt single-end reads, resulting in 7,221,225 and 6,129,038 reads for Luc KD and 7,025,416 and 5,763,089 reads for Spn-E KD (Replicates 1 and 2, respectively).

## Total RNA sequencing

BmN4 cells treated with Luc (control) or Siwi siRNA were used for RNA sequencing. Total RNAs were isolated using ISOGEN II (FUJIFILM Wako Pure Chemical) and DNase (Life Technologies). RNA libraries were prepared using the NEBNext Ultra II Directional RNA Library Prep Kit for Illumina (New England Biolabs) with the Ribo-Zero Plus rRNA Depletion Kit (Human/Mouse/Rat) (Illumina) and sequenced on the NovaSeq platform (Illumina) to obtain paired-end reads, resulting in 77,614,516 and 68,702,400 reads for Luc KD and 81,135,072 and 77,184,922 reads for Siwi KD (Replicates 1 and 2, respectively).

## Mapping of FAST-iCLIP reads

After removal of 3′ adaptor sequences by Cutadapt (version 4.0)[57], reads shorter than 27 nt were discarded. PCR duplicates were removed by collapsing all identical reads containing the same random barcode at the 5′-end using fastx-collapser in FASTX-Toolkit (version 0.0.14) (http://hannonlab.cshl.edu/fastx_toolkit/index.html); then, the 5′ random barcode was removed from the reads. The reads were mapped against the *Bombyx mori* genome assembly (Nov. 2016) from the Silk-Base database using STAR (version 2.7.9a)[58] in random and multiple counting modes allowing up to 4% mismatches relative to read length. Genome-mapped reads were remapped against the silkworm gene library (see "Gene classification" section) using Bowtie (version 1.3.1)[59] in random and multiple counting modes allowing up to two mismatches.

## Mapping of total RNA sequencing reads

After cleaning raw reads and removal of adaptor sequences using Rcorrector (version 1.0.4)[60], TranscriptomeAssemblyTools (https://github.com/harvardinformatics/TranscriptomeAssemblyTools), and

TrimGalore (version 0.6.6) (https://github.com/FelixKrueger/TrimGalore), reads shorter than 36 nt were discarded. The reads were mapped against the genome and remapped against the silkworm gene library using STAR in random and multiple counting modes allowing up to 4% mismatches relative to read length.

## Read count calculation

Gene-mapped reads from the FAST-iCLIP and the total RNA sequencing data were processed using SAMtools (version 1.15)[61]. The reads mapped to the sense direction were normalized against the number of genome-mapped reads to RPM or RPKM with the addition of a pseudo-count for log transformation and FC calculation. Owing to the high correlation of replicate samples, the two replicate reads were merged for further analysis. A total of 3006 sequences with >10 RPKM in the Luc KD (control) or Siwi KD RNA sequencing data, which were classified into protein-coding genes (2617 sequences) or transposons (389 sequences), were focused on in the subsequent analyses.

## Transposon classification

To reference as reliable annotations for transposons, the clades of 1716 *Bombyx mori* transposons[31] were used. A total of 100 sequences with a robust annotation were extracted from 389 sequences and classified into five clades: LINE (25 sequences), SINE (8 sequences), LTR (33 sequences), DNA (33 sequences), and DIRS (1 sequence). Of these, the DIRS clade was excluded from the analysis because it included too few sequences.

## Statistics and reproducibility

Immunoprecipitation of Supplementary Fig. 1e was performed over six independent experiments, and other immunoprecipitation, rescue assay, CLIP, immunofluorescence, live cell imaging, in vitro droplet formation assay, electrophoretic mobility shift assay, FAST-iCLIP, and total RNA sequencing were performed over two independent experiments. Statistical analyses and data visualization were performed in the R software environment (version 4.2.0) or Python 3 with add-on libraries (Numpy, Scipy, Pandas, and Matplotlib). The statistical methods are indicated in the legends. No statistical methods were used to predetermine sample size. The experiments were not randomized, and investigators were not blinded to allocation during the experiments and outcome assessment.

## Reporting summary

Further information on research design is available in the Nature Portfolio Reporting Summary linked to this article.

## Data availability

Source data are provided with this paper. The data supporting the findings of this study are available from the corresponding author upon reasonable request. The FAST-iCLIP and total RNA sequencing data generated in this study have been deposited in Gene Expression Omnibus under accession code GSE213917. The silkworm genome databases used in this study (KAIKObase database[53] and Silkbase database[54]) are available under the following links: [https://sgp.dna.affrc.go.jp/KAIKObase/, https://silkbase.ab.a.u-tokyo.ac.jp/cgi-bin/index.cgi]. Source data are provided with this paper.

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

## Acknowledgements
We thank K. Sato, S. Hirakata, and Y.W. Iwasaki for useful discussions. We also thank R. Murakami and T. Sumiyoshi for technical help and R. Pillai for sharing the BmVasa RK mutant plasmid. This study was supported by MEXT KAKENHI Grant Number JP19H05466 (to M.C.S.); JSPS KAKENHI Grant Numbers JP21K15039 (to H.Y.), JP16J02082 (to Y.N.), and JP20K06483 (to K.M.N.); and the Sumitomo Foundation Grant for Basic Science Research Project (to H.Y.).

## Author contributions
H.Y., Y.N., S.K., K.M.N., and A.K. performed biochemical experiments. Y.N. performed bioinformatic analyses. M.C.S. supervised and discussed the work and wrote the manuscript with the other authors.

## Competing interests
The authors declare no competing interests.

## Additional information

**Peer review information** : *Nature Communications* thanks Paul Lasko and the other anonymous reviewer(s) for their contribution to the peer review of this work. Peer reviewer reports are available.

