## [Peer Review File · Nature Communications]

Bombyx Vasa sequesters transposon mRNAs in nuage via phase separation requiring RNA binding and self-associationREVIEWER COMMENTS

Reviewer #1 (Remarks to the Author):

In this study Yamazaki et al. identify the essential components of BmVasa protein in Vasa body assembly via phase separation. The authors provide compelling evidence for cooperation of the N-terminal IDR (N-IDR) and helicase domain and suggest the phase separation of BmVasa is responsible for transposon mRNAs sequestration in nuage, which in turn leads to the Siwi-dependent transposon repression and Ago3-piRISC biogenesis. Experimental and computational approaches are appropriate and the reporting of the methodology is generally sufficient. The data presented in the manuscript are very robust and support the main conclusions of the authors, however a few additional experiments might be needed for the acceptance of this manuscript for publication in Nature Communications. Therefore, the manuscript requires a revision addressing the following comments and suggestions:

1. The authors used various BmVasa mutant constructs for in-vivo and in-vitro droplet formation assays with the outcomes summarized in Supp. Fig. 1f.

The N-E-C construct, which was used to mimic the Ddx4-YFP experiments reported by others, does not localize to the Vasa bodies in-vivo. Perhaps, this can be explained by the expression levels that are not enough to induce phase separation hence are not detectable as droplets by immunofluorescence experiments. Could the authors show expression levels of this construct to overrule this possibility? And to follow up; they might try to push expression levels of this construct for instance, by utilizing a stronger promoter to drive the expression?

The R470A mutant appears to form peri-nucleolar bodies in the nucleus and it generally looks more nuclear than the other constructs, could the authors comment on this? In addition, self association of this FLAG-R470A mutant with myc-WT construct does seem to be affected (Supp. Fig.1d), unlike the text referring to this result.

2. In Fig.2g Δ N mutant used in in-vitro droplet formation assays indicate no detectable droplets, however only a single concentration (1.5 μ M) of the recombinant protein was used in these assays. The authors could also test higher concentrations to show either this mutant is incapable of LLPS or it forms droplets at higher concentrations.

3. The in-vitro droplet formation assays performed in the presence of RNA are indicated to have single stranded RNA (ssRNA). According to the methods section this RNA is in-vitro transcribed from a plasmid (pAcM-lacZ) and a 2000nt long RNA can hardly be devoid of secondary structures. Nevertheless, considering the piRNA pathway with the involvement of the double stranded RNA at certain stages, carrying out these experiments with another 50nt RNA species in sense, antisense orientation and also designed-to-be double stranded version would be informative.

4. The authors concluded that the sDMA modifications are not important for LLPS, but they do not overrule the alterations in kinetics of events. The authors could perform a FRAP experiment to show the RK mutant had altered recovery or not (same as done for E339Q mutant).

5. The authors employ FAST-iCLIP for BmVasa target determination. According to the methods section of the manuscript, the method is very similar to iCLIP however in the absence of a schematic drawing of the protocol or a bench-marking by comparing to existing methodologies it is difficult to judge the data generated by this method. Moreover, there is not a single genome browser view of the genome wide dataset generated (e.g. on piRNA

clusters) or access to the GEO database submission. The authors should provide the necessary information to the reviewers.

6. In FAST-iCLIP experiments, the authors probably should've depleted Siwi for the BmVasa, since UV-irradiation used in CLIP experiments is quite mild (100-200mJ/cm²) which unlikely to cause any DNA damage.

7. One of the major questions of this work, which is "do the Vasa bodies form upon binding to RNA or do the formation of Vasa bodies enable RNA-binding?", still remains to be tested. Perhaps an in-vitro radioactive CLIP assay, similar to the one presented in Fig. 1i could be performed with wt, vs ΔN vs R470A, E339Q by mixing them with RNA. The results of this experiment could boost the conclusions of this study.

A few minor points:

1. Supplementary Figure 1 has only panels a) to f) but the panels a) and b) are not labeled on the figure.

2. In Line 214 the cited reference #38 is a review and is not showing the importance of DDX4 sDMA in LLPS. Could this be a mistake and the authors actually wanted to cite reference #32?

3. In Fig. 3 the panel orientation needs to be optimized as the quantification of the results in panel b) might be confused for the last image of panel a) (that is 2000nt ssRNA mixing experiment). In addition the quantification plots (Number / 1mm²) should be labeled properly with WT and R470A respectively.

4. A brief comment about UAP56: it is a general EJC factor, however past work of Brennecke group (IMBA, Vienna) has identified a specialized export factor for piRNA precursors; Nxf3. Therefore, this information in the introduction needs to be updated accordingly.

Reviewer #2 (Remarks to the Author):

The manuscript by Yamazaki et al reports the results of their investigation of the Vasa protein of *Bombyx mori* (BmVasa), focusing on how it undergoes phase separation in vitro, how it assembles nuage-like structures (Vasa bodies) in vivo, and its associated RNAs. Its major finding is that, unlike for other Vasa orthologues, the N-terminal internally disordered region (N-IDR) of BmVasa is insufficient for phase separation in vitro and for Vasa body formation in vivo. For BmVasa the RNA helicase domain is also essential for these functions. They also show that the N-IDR is essential for full RNA binding activity. Finally, they show that BmVasa binds preferentially to transposon RNAs, whether or not Siwi activity is present.

While these results are interesting, my opinion is that the scientific advance documented here is not of sufficient significance to warrant publication in a broad-interest journal such as Nature Communications. The manuscript documents differences between BmVasa and previously studied orthologues, but does not provide a solid mechanistic picture of why these differences arise.

The Discussion and Supplemental Figure 5 proposes that differences in the number of Phe residues in the N-IDRs of BmVasa and mammalian orthologues underlie their different properties. The paper would be strengthened if this idea were experimentally tested, by analyzing (at least in vitro) chimeric proteins that fuse the N-IDR of one orthologue with the helicase domain of another, or by using site-specific mutagenesis to test the effects of adding or subtracting Phe residues. Supplemental Figure 5 and the mechanistic analysis could usefully be expanded to include *D. melanogaster* Vasa, which is intermediate in properties to those shown. Vasa N-IDRs evolve rapidly and vary greatly even among closely

related species where one might imagine they function similarly. In this context it would be interesting to look experimentally at *D. pseudoobscura* Vasa, which has a relatively short N-IDR that contains only one Phe residue.

Other comments:

1. With regard to Fig 3, given that BmVasa is an RNA helicase, might the differential effects of adding 2000 nt long ssRNA rather than 50 nt long ssRNA result from the presence of more extensive secondary structures in the longer RNA, instead of simply resulting from the different lengths? I also find this figure to be poorly organized. The panel that shows data from R470A in the presence of 2000 nt ssRNA is next to an image of WT in the presence of 50 nt ssRNA, making comparison difficult. Also the negative data with R470A in the presence of 50 nt ssRNA should be shown.

2. Measures of statistical significance underlying the conclusions drawn from the data in Fig 4 are absent, and should be provided.

3. The labelling of panels in Fig 1 does not correspond to their description in the text. The IP experiment is Fig 1b in the text but 1d in the figure itself. As a result, panels 1b and 1c in the figure correspond to 1c and 1d in the text.

4. The narrative contains numerous instances of awkward phrasing and a few minor English presentational errors that should be corrected by copyediting. The introduction could also be shortened somewhat.

Reviewer #3 (Remarks to the Author):

The authors describe that the silkworm BmVasa WT protein and a construct lacking the C-terminal unstructured part (Δ C) restore nuage formation in KO cells, whereas a construct lacking the N-terminal unstructured part (Δ N) cannot. Cells expressing the Δ N mutant also fail to assemble functional RISC complexes. Self-association is not sufficient for vasa body formation though, since a mutant that presumably abolishes RNA binding also does not rescue Vasa body formation. Recombinant wild-type BmVasa forms condensate-like structures in vitro, however in the presence of high amounts (15%) PEG, which is reduced to 5% in the presence of long ssRNA. The Δ N mutant fails to form condensates, whereas the R470A mutant still does. Both in cells and in vitro, condensate structures are sensitive to ammonium acetate. Addition of ATP increases condensate formation for a catalytically inactive mutant. Furthermore, the authors map which RNAs (mRNAs / transposons) bind to BmVasa by iClip experiments.

I feel that the experiments describing condensate formation are not novel enough, and not sufficiently linked to the very promising FAST-iCLIP experiments, to justify publication in this journal.

Specific comments

- the labelling for the IF figures is hard to read, it would be helpful to write e.g. "Flag-WT" or "Flag-N-EGFP-C" in Fig 1b/f etc.

- please quantify expression levels of your FLAG constructs for the IF in Figure 1; they can substantially influence formation of condensate structures.

- it is claimed that the N-IDR does not have intrinsic RNA binding; however this is not directly tested, the R470A mutant in Fig 1G does show some residual RNA binding.

- please track the the condensates for a longer time-frame in Fig 2A, from the FRAP experiments - which hint that these structures have relatively little turnover - it is surprising that they would fuse so readily

- figures of in vitro condensates: please explain how your condensates were quantified (i.e. the histograms) in more detail, it is hard to understand from the figure legend. You are showing very few condensates, and without proper quantification, the effects you describe are hard to follow / judge, or to compare (e.g. Fig 3c)

- why do condensates in Fig 2e look different (i.e. larger (? might be a different scale bar) / less intense / less round) from e.g. figure 2 d/f ?

- figure 3a: please label with more detail; this is all wild type protein if I understand correctly ?

- DQAD mutants typically have no detectable ATPase activity, so I suspect the protein remains bound to ATP. Also, most DEAD-box proteins require ATP to bind RNA at the canonical RecA core, which should be discussed accordingly with respect to the experiment in Figure 3C.

- Figure 4: this data seems disconnected from the rest of the story. It would have been very interesting to see the effect of the ΔN or R470A mutant for these experiments, to link it back to Figure 1.

Point-by-point responses to reviewers' comments

Reviewer #1

1. The authors used various BmVasa mutant constructs for in-vivo and in-vitro droplet formation assays with the outcomes summarized in Supp. Fig. 1f.

The N-E-C construct, which was used to mimic the Ddx4-YFP experiments reported by others, does not localize to the Vasa bodies in-vivo. Perhaps, this can be explained by the expression levels that are not enough to induce phase separation hence are not detectable as droplets by immunofluorescence experiments. Could the authors show expression levels of this construct to overrule this possibility? And to follow up; they might try to push expression levels of this construct for instance, by utilizing a stronger promoter to drive the expression?

We thank the reviewer for pointing this out. We repeated the experiments where we measured the expression levels of WT BmVasa and its N-EGFP-C mutant (initially N-E-C) in individual cells and examined Vasa body localization of the proteins in these cells. WT localized to Vasa bodies even at lower levels, whereas N-EGFP-C failed to do so even at higher levels. Representative cells are shown in revised Supplementary Fig. 1b. In this series of experiments, a strong promoter (i.e., OpIE2 promoter) was originally used; therefore, increasing the expression levels by changing the promoter would be difficult to achieve.

The R470A mutant appears to form peri-nucleolar bodies in the nucleus and it generally looks more nuclear than the other constructs, could the authors comment on this? In addition, self association of this FLAG-R470A mutant with myc-WT construct does seem to be affected (Supp. Fig.1d), unlike the text referring to this result.

We thank the reviewer for pointing this out. Indeed, the R470A mutant appeared at low levels around peri-nucleolar regions when endogenous BmVasa was absent from the cells. In plants, chromosome regions around the nucleolus, namely, nucleolus-associated chromatin domains (NADs), are rich in transposons, which could be expressed although they may not be abundant ([10.1016/j.celrep.2016.07.016](https://doi.org/10.1016/j.celrep.2016.07.016)). Thus, one interesting idea is that BmVasa visits the peri-nucleolar regions, binds to transposon mRNAs, and returns to the cytoplasm to initiate assembly of Vasa bodies with those RNAs. Some fraction of the R470A mutant appears to remain there, likely due to the lack of RNA-binding activity. However, this is only speculative, and we are even uncertain whether NADs exist in BmN4 cells. We will explore this interesting possibility and hopefully publish the outcome of this exploration in a forthcoming paper. To respond to

the reviewer's suggestion, we added a sentence in the revised text (page 8), "*It is noted that the R470A mutant often appeared to be peri-nucleolar when endogenous BmVasa was absent, although its physiological significance remains unknown.*" This is just additional information, but DmVasa was reported to localize to some type of nuclear structure in primordial germ cells (10.7554/eLife.37949), although its physiological significance remains unknown. Regarding the self-association data in original Supplementary Fig. 1d; we repeated the experiments six times and quantified the results, which are shown as revised Supplementary Fig. 1e. The gel image was replaced with a representative one from the new experiments.

2. In Fig.2g ΔN mutant used in in-vitro droplet formation assays indicate no detectable droplets, however only a single concentration (1.5 μM) of the recombinant protein was used in these assays. The authors could also test higher concentrations to show either this mutant is incapable of LLPS or it forms droplets at higher concentrations.

We thank the reviewer for the suggestions. We performed the experiments with higher concentrations of WT BmVasa and its ΔN mutant. The results are shown in revised Supplementary Fig. 2e.

3. The in-vitro droplet formation assays performed in the presence of RNA are indicated to have single stranded RNA (ssRNA). According to the methods section this RNA is in-vitro transcribed from a plasmid (pAcM-lacZ) and a 2000nt long RNA can hardly be devoid of secondary structures. Nevertheless, considering the piRNA pathway with the involvement of the double stranded RNA at certain stages, carrying out these experiments with another 50nt RNA species in sense, antisense orientation and also designed-to-be double stranded version would be informative.

We thank the reviewer for the suggestions. We performed the assays using two types of 50 nt ssRNAs (lacZ and BMgn003210) and a 50 nt dsRNA (lacZ). In each case, BmVasa failed to assemble droplets. The results are indicated in revised Supplementary Fig. 3b.

4. The authors concluded that the sDMA modifications are not important for LLPS, but they do not overrule the alterations in kinetics of events. The authors could perform a FRAP experiment to show the RK mutant had altered recovery or not (same as done for E339Q mutant).

We thank the reviewer for pointing this out. We performed FRAP experiments with WT BmVasa and its RK mutant. The results are indicated in revised Supplementary Fig. 2c.

It is noted that the experiments were performed *in vivo* because *E. coli*-made recombinant BmVasa does not have sDMA modification.

5. The authors employ FAST-iCLIP for BmVasa target determination. According to the methods section of the manuscript, the method is very similar to iCLIP however in the absence of a schematic drawing of the protocol or a bench-marking by comparing to existing methodologies it is difficult to judge the data generated by this method. Moreover, there is not a single genome browser view of the genome wide dataset generated (e.g. on piRNA clusters) or access to the GEO database submission. The authors should provide the necessary information to the reviewers.

We thank the reviewer for pointing this out. The description of the FAST-iCLIP method was modified in the revised text (page 29) to clarify the difference from the iCLIP method. The FAST-iCLIP and total RNA sequencing data were deposited in the GEO database and the accession number was indicated in the revised text (page 33). To access the data in the review process, please go to <https://www.ncbi.nlm.nih.gov/geo/query/acc.cgi?acc=GSE213917> and enter code "cjulccksvdgpvcx" in the box.

6. *In FAST-iCLIP experiments, the authors probably should've depleted Siwi for the BmVasa, since UV-irradiation used in CLIP experiments is quite mild (100-200mJ/cm²) which unlikely to cause any DNA damage.*

We thank the reviewer for pointing this out. We have attempted more than a few times to generate FAST-iCLIP libraries from Siwi-depleted cells, but unfortunately were not successful. Spn-E knockdown depletes Siwi-piRISC as efficiently as the loss of Siwi, as was noted in the original text (page 13). Because it was unclear whether UV irradiation was the cause of the trouble, the relevant sentence in the revised text (page 15) was changed to read, "..... *because Siwi-depleted conditions did not yield a high-quality library.*"

7. *One of the major questions of this work, which is "do the Vasa bodies form upon binding to RNA or do the formation of Vasa bodies enable RNA-binding?", still remains to be tested. Perhaps an in-vitro radioactive CLIP assay, similar to the one presented in Fig.1i could be performed with wt, vs ΔN vs R470A, E339Q by mixing them with RNA. The results of this experiment could boost the conclusions of this study.*

We thank the reviewer for pointing this out. The R470A mutant failed to assemble Vasa

bodies *in vivo*, even though it contains a complete N-IDR. Namely, without RNA binding via the RNA helicase domain, Vasa bodies cannot be assembled in the first place. Thus, the second scenario "Vasa body formation enables RNA-binding" is ruled out.

A few minor points:

1. Supplementary Figure 1 has only panels a) to f) but the panels a) and b) are not labeled on the figure.

We fixed this problem.

2. In Line 214 the cited reference #38 is a review and is not showing the importance of DDX4 sDMA in LLPS. Could this be a mistake and the authors actually wanted to cite reference #32?

We thank the reviewer for pointing this out. To make the section easier to read, the positions of several sentences have been swapped. In this process, Ref. 38 was removed.

3. In Fig.3 the panel orientation needs to be optimized as the quantification of the results in panel b) might be confused for the last image of panel a) (that is 2000nt ssRNA mixing experiment). In addition the quantification plots (Number / 1mm²) should be labeled properly with WT and R470A respectively.

We fixed these problems.

4. A brief comment about UAP56: it is a general EJC factor, however past work of Brennecke group (IMBA, Vienna) has identified a specialized export factor for piRNA precursors; Nxf3. Therefore, this information in the introduction needs to be updated accordingly.

We updated the information accordingly in the revised text (page 3).

Reviewer #2

The Discussion and Supplemental Figure 5 proposes that differences in the number of Phe residues in the N-IDRs of BmVasa and mammalian orthologues underlie their different properties. The paper would be strengthened if this idea were experimentally tested, by analyzing (at least in vitro) chimeric proteins that fuse the N-IDR of one orthologue with the helicase domain of another, or by using site-specific mutagenesis to test the effects of adding or subtracting Phe residues. Supplemental Figure 5 and the

mechanistic analysis could usefully be expanded to include D. melanogaster Vasa, which is intermediate in properties to those shown. Vasa N-IDRs evolve rapidly and vary greatly even among closely related species where one might imagine they function similarly. In this context it would be interesting to look experimentally at D. pseudoobscura Vasa, which has a relatively short N-IDR that contains only one Phe residue.

We thank the reviewer for the suggestions. We added DmVasa in revised Fig. 4a and 4b. Instead, MVH was removed from the figures to keep the section concise. We also fluorescently examined the cellular localization of N-EGFP-C (initially N-E-C) and its mutants, where BmVasa N-IDR was replaced with DmVasa N-IDR [N(DmVasa)-EGFP-C] or Ddx4 N-IDR [N(Ddx4)-EGFP-C]. N(Ddx4)-EGFP-C, but not N(DmVasa)-EGFP-C, assembled nuage-like structures as was previously reported for the Ddx4^{YFP} mutant [i.e., N(Ddx4)-YFP-C(Ddx4)] in HeLa cells (10.1016/j.molcel.2015.01.013). Intriguingly, when all 14 phenylalanines of N-IDR in N(Ddx4)-EGFP-C were replaced with alanine, the mutant no longer assembled nuage-like structures. These results are shown in revised Fig. 4c, which clearly suggests that Ddx4 N-IDR alone has the capacity to induce LLPS-mediated nuage-like structure formation *in vivo*, and that phenylalanines in the region are the key to allowing the N-IDR to reach the full potential. In the revised manuscript, this section was moved to the main text as “*Peptide composition of N-IDR influences the ability of Vasa to assemble nuage-like structures*” (page 13 in the revised text). Accordingly, the Discussion section was also modified in the revised text (pages 18-21). It is noted that the nuclear immunofluorescent signal of Flag-N-EGFP-C by the anti-Flag antibody (revised Supplementary Fig. 1b) appeared slightly weaker than the green fluorescent signal (revised Fig. 4c). This may reflect the structure of this mutant in the nucleus: i.e., the N-terminus may not be sufficiently accessible to antibodies or other molecules. Importantly, no nuage-like structures were detected in either case. Thus, this has no impact on the interpretation of this study. The Vasa of *D. pseudoobscura* was not experimentally examined in this study as we considered that it is beyond the scope of this work. However, we thank the reviewer for raising this interesting issue and are planning to investigate Vasa of *D. pseudoobscura* along with that of other *Drosophila* relatives as a separate study.

Other comments:

1. With regard to Fig 3, given that BmVasa is an RNA helicase, might the differential effects of adding 2000 nt long ssRNA rather than 50 nt long ssRNA result from the presence of more extensive secondary structures in the longer RNA, instead of simply

resulting from the different lengths? I also find this figure to be poorly organized. The panel that shows data from R470A in the presence of 2000 nt ssRNA is next to an image of WT in the presence of 50 nt ssRNA, making comparison difficult. Also the negative data with R470A in the presence of 50 nt ssRNA should be shown.

We thank the reviewer for the suggestions. We rearranged the panels in Fig. 3a. The negative data with R470A in the presence of 50 nt ssRNA were included in the figure. We performed the assays with several other 2,000 nt ssRNAs, whose sequences were distinct from the sequence of original 2,000 nt ssRNA (2,000 nt lacZ). In each case, the droplets were similarly assembled. We added the results of 2,000 nt of *Bombyx mori* protein-coding gene, BMgn003210, as a representative in revised Supplementary Fig. 3b.

2. Measures of statistical significance underlying the conclusions drawn from the data in Fig 4 are absent, and should be provided.

Measures of statistical significance in original Fig. 4 are provided in the revised manuscript (Fig. 5). Also, we reclassified 389 transposons into three groups according to the degree of repression caused by the lack of Siwi-piRISC (revised Fig. 5d) and analyzed them. The results showed that the binding of BmVasa to transposon mRNAs was not significantly affected by the degree of repression (revised Fig. 5d). We also analyzed transposon and protein-coding mRNAs in total and BmVasa binding by focusing on their length. The results (revised Fig. 5e) showed that BmVasa binds to transposon mRNAs with little dependence on their length, unlike protein-coding mRNAs.

3. The labelling of panels in Fig 1 does not correspond to their description in the text. The IP experiment is Fig 1b in the text but 1d in the figure itself. As a result, panels 1b and 1c in the figure correspond to 1c and 1d in the text.

We checked and corrected the figure labeling throughout the revised manuscript.

4. The narrative contains numerous instances of awkward phrasing and a few minor English presentational errors that should be corrected by copyediting. The introduction could also be shortened somewhat.

The English of the manuscript was proofread by a native English-speaking expert. We tried to shorten the Introduction, but we realized that all pieces of information there are important for readers, so the Introduction essentially remains as it was.

Reviewer #3

- the labelling for the IF figures is hard to read, it would be helpful to write e.g. "Flag-WT" or "Flag-N-EGFP-C" in Fig 1b/f etc.

We thank the reviewer for the suggestions. We changed F-WT and F-N-E-C to Flag-WT and Flag-N-EGFP-C, respectively. Similar changes were made throughout the revised manuscript.

- please quantify expression levels of your FLAG constructs for the IF in Figure 1; they can substantially influence formation of condensate structures.

We thank the reviewer for this comment. To ensure that the analysis is accurate, we expressed WT BmVasa and its mutants, Δ N, Δ C, N-EGFP-C (originally N-E-C), and R470A, individually in BmN4 cells but in parallel, and analyzed them. WT BmVasa and its Δ C mutant were localized to Vasa bodies when expressed similarly to the levels of Δ N, N-EGFP-C, and R470A, which did not localize to Vasa bodies at those levels. Vasa body localization of WT and Δ C mutant was observed even when less abundantly expressed. Vasa body localization of Δ N, N-EGFP-C, and R470A mutants was hardly observed even at higher levels. The results are shown in revised Supplementary Fig. 1b.

- it is claimed that the N-IDR does not have intrinsic RNA binding; however this is not directly tested, the R470A mutant in Fig 1G does show some residual RNA binding.

We thank the reviewer for pointing this out. To directly test the RNA-binding capacity of N-IDR, we performed electronic mobility shift assays. The results are shown in revised Supplementary Fig. 1f. While a strong shift was observed with WT BmVasa, the N-IDR exhibited very low RNA-binding capacity. Accordingly, we changed the original sentence to read, "*N-IDR may have minimal affinity for RNA but is clearly insufficient to confer full RNA-binding activity of BmVasa. Rather, N-IDR supports the central RNA helicase domain to confer full RNA-binding activity of BmVasa via self-association.*" (page 8 in the revised text).

- please track the the condensates for a longer time-frame in Fig 2A, from the FRAP experiments - which hint that these structures have relatively little turnover - it is surprising that they would fuse so readily

We thank the reviewer for this suggestion. We show the longer time-frame data as Supplementary Movie 1.

- figures of in vitro condensates: please explain how your condensates were quantified (i.e. the histograms) in more detail, it is hard to understand from the figure legend. You are showing very few condensates, and without proper quantification, the effects you describe are hard to follow / judge, or to compare (e.g. Fig 3c)

We thank the reviewer for pointing this out. We described in more detail the quantification of condensates in revised Methods (pages 27-28). We also included representative uncropped images as revised Supplementary Fig. 2d, 3a, and 3c.

- why do condensates in Fig 2e look different (i.e. larger (? might be a different scale bar) / less intense / less round) from e.g. figure 2 d/f ?

We thank the reviewer for pointing this out. The experiment was repeated, and the new results are presented as revised Fig. 2e.

- figure 3a: please label with more detail; this is all wild type protein if I understand correctly ?

We fixed this problem.

- DQAD mutants typically have no detectable ATPase activity, so I suspect the protein remains bound to ATP. Also, most DEAD-box proteins require ATP to bind RNA at the canonical RecA core, which should be discussed accordingly with respect to the experiment in Figure 3C.

As initially noted in the original text (page 11), the BmVasa DQAD (E399A) mutant hydrolyzes ATP to ADP and inorganic phosphate, but both small molecules remain bound to the protein along with the bound RNA (Ref. 27). To further clarify this section, we added a sentence to the revised text (page 12), “*This indicates that the droplet-forming capacity of the E399Q mutant is stronger than that of the WT.*”

- Figure 4: this data seems disconnected from the rest of the story. It would have been very interesting to see the effect of the ΔN or R470A mutant for these experiments, to link it back to Figure 1.

We thank the reviewer for pointing this out. The RNA-binding activity of both ΔN and R470A mutants was so weak that we could not perform the suggested experiments. After discovering that BmVasa self-associates (via N-IDR) and binds to RNA (via RNA helicase domain with indispensable support from N-IDR) to assemble nuage, the Ago3-piRISC generation center, via phase separation, it is a very logical and natural development that we became interested in the question of what RNAs BmVasa actually

binds to. We found that BmVasa selectively binds to transposon mRNAs. In the revised manuscript, we also clarified that this tendency is largely independent of RNA length, abundance, or simple primary sequence. Finally, we propose a functional model for BmVasa in the formation of Vasa bodies and Ago3-piRISC generation in silkworm germ cells (revised Fig. 6). Thus, we state that all experiments conducted in this study were valuable and well connected with each other.

REVIEWERS' COMMENTS:

Reviewer #1 (Remarks to the Author):

The revised version of the manuscript by Yamazaki et al. contains new results presented coherently with the earlier results. Especially, the new Figure 4 with the effect of the N-IDR peptide composition on the nuage-like structures with an evolutionary view gave new insights into important residues needed for the assembly of these structures. Moreover, re-wording in some sections as well as additional clarifications of earlier results and alterations in the presentation of the data significantly enhanced the manuscript. The introduction section is adequate as it provides a succinct view on the state of the art in various species and perfectly sets the ground for the reasons of studying Vasa in *Bombyx mori*. In brief; the authors have substantially improved the study in the revised version. All my points were very nicely addressed. I do not have any further suggestions, thus recommend this manuscript to be accepted in Nature Communications.

Reviewer #2 (Remarks to the Author):

The major concern I expressed in my previous review has been addressed in the revised version, with the addition of new experimental data investigating the *D. melanogaster* N-IDR and the effects of altering all Phe residues in the human Ddx4 N-IDR.

These new data support the paper's conclusions. However, they also raise the question of what compensatory mechanisms there are to enable orthologous Vasa proteins with different N-IDR functions to nevertheless have similar functions in nuage assembly and the piRNA pathway. This issue is mentioned in the discussion (lines 428-442) but left as an unanswered question. Similarly, no specific model is provided as to how BmVasa binding might be sensitive to RNA length for protein-coding RNAs, but not so for transposon RNAs.

For me, therefore, while the paper is certainly sound, its results and the conclusions drawn from them do not represent a major conceptual advance from what is presently established in the field.

Reviewer #3 (Remarks to the Author):

The authors addressed all my comments sufficiently.

Point-by-point responses to Reviewers' comments

Reviewer #1

The revised version of the manuscript by Yamazaki et al. contains new results presented coherently with the earlier results. Especially, the new Figure 4 with the effect of the N-IDR peptide composition on the nuage-like structures with an evolutionary view gave new insights into important residues needed for the assembly of these structures. Moreover, re-wording in some sections as well as additional clarifications of earlier results and alterations in the presentation of the data significantly enhanced the manuscript. The introduction section is adequate as it provides a succinct view on the state of the art in various species and perfectly sets the ground for the reasons of studying Vasa in Bombyx mori. In brief; the authors have substantially improved the study in the revised version. All my points were very nicely addressed. I do not have any further suggestions, thus recommend this manuscript to be accepted in Nature Communications.

We thank this reviewer for noting that the revisions we have made to the original manuscript were satisfactory.

Reviewer #2

The major concern I expressed in my previous review has been addressed in the revised version, with the addition of new experimental data investigating the D. melanogaster N-IDR and the effects of altering all Phe residues in the human Ddx4 N-IDR. These new data support the paper's conclusions.

We thank this reviewer for noting that the revisions we have made to the original manuscript were satisfactory.

However, they also raise the question of what compensatory mechanisms there are to enable orthologous Vasa proteins with different N-IDR functions to nevertheless have similar functions in nuage assembly and the piRNA pathway. This issue is mentioned in the discussion (lines 428-442) but left as an unanswered question. Similarly, no specific model is provided as to how BmVasa binding might be sensitive to RNA length for protein-coding RNAs, but not so for transposon RNAs. For me, therefore, while the paper is certainly sound, its results and the conclusions drawn from them do not represent a major conceptual advance from what is presently established in the field.

We appreciate this reviewer's comments. Understanding the compensatory mechanisms that allow Vasa orthologs with different N-IDR functions to perform similar functions in

the nuage formation and piRNA pathway is indeed an important challenge. To solve this question, further experiments, including a comprehensive analysis of factors that bind to Vasa, are underway. We hope that these experiments will lead to new discoveries in the future.

Regarding BmVasa binding being sensitive to RNA length (and RNA abundance) for protein-coding mRNAs but not for transposon mRNAs: Our *in vitro* assays showed that BmVasa tends to assemble droplets when longer RNAs are present. This is probably true *in vivo* as well. In the case of transposon mRNAs, however, such a tendency was not evident. As noted in the Discussion section (page 21), we speculate that “factors that bind to BmVasa may aid in the selectivity (for transposon mRNAs)”. As noted in the Discussion section (page 21), we are currently aiming to identify factors that bind to Vasa. We hope that this experiment will lead to new discoveries in the future.

Reviewer #3

The authors addressed all my comments sufficiently.

We thank this reviewer for noting that the revisions we have made to the original manuscript were satisfactory.